# Both abundant and rare fungi colonizing *Fagus sylvatica* ectomycorrhizal root-tips shape associated bacterial communities

Marlies Dietrich[1], Alicia Montesinos-Navarro [2], Raphael Gabriel[1,4], Florian Strasser[1], Dimitri V. Meier[1,5], Werner Mayerhofer[1], Stefan Gorka[1], Julia Wiesenbauer[1], Victoria Martin[1], Marieluise Weidinger[3], Andreas Richter[1], Christina Kaiser [1✉] & Dagmar Woebken [1✉]

Ectomycorrhizal fungi live in close association with their host plants and form complex interactions with bacterial/archaeal communities in soil. We investigated whether abundant or rare ectomycorrhizal fungi on root-tips of young beech trees (*Fagus sylvatica*) shape bacterial/archaeal communities. We sequenced 16S rRNA genes and fungal internal transcribed spacer regions of individual root-tips and used ecological networks to detect the tendency of certain assemblies of fungal and bacterial/archaeal taxa to inhabit the same root-tip (i.e. modularity). Individual ectomycorrhizal root-tips hosted distinct fungal communities associated with unique bacterial/archaeal communities. The structure of the fungal-bacterial/archaeal association was determined by both, dominant and rare fungi. Integrating our data in a conceptual framework suggests that the effect of rare fungi on the bacterial/archaeal communities of ectomycorrhizal root-tips contributes to assemblages of bacteria/archaea on root-tips. This highlights the potential impact of complex fine-scale interactions between root-tip associated fungi and other soil microorganisms for the ectomycorrhizal symbiosis.

[1] Department of Microbiology and Ecosystem Science, Centre for Microbiology and Environmental Systems Science, University of Vienna, Vienna, Austria. [2] Centro de Investigaciones Sobre Desertificación (CIDE, CSIC-UV-GV), Carretera de Moncada-Náquera Km 4.5, 46113 Moncada, Valencia, Spain. [3] Core Facility Cell Imaging and Ultrastructure Research, University of Vienna, Vienna, Austria. [4] Present address: Solugen Inc., 14549 Minetta St, Houston, TX 77035, USA. [5] Present address: Institute of Biogeochemistry and Pollutant Dynamics, Swiss Federal Institute of Technology, Zurich (ETH Zurich), Zurich, Switzerland. ✉email: christina.kaiser@univie.ac.at; dagmar.woebken@univie.ac.at

The mycorrhizal symbiosis is an ancient association between plants and soil fungi, which facilitates the growth of both partners by transferring benefits, such as reciprocal nutrient acquisition, increased stress resistance and protection from pathogens[1,2]. Fungal hyphae emanating from mycorrhizal root-tips expand the surface area of the host root system for enhanced nutrient uptake and potential tripartite interactions with other soil microorganisms[3]. The mycorrhizosphere, i.e., the soil in direct contact with mycorrhizal hyphae influenced by root and hyphal exudates, provides an attractive habitat for soil bacteria and archaea. The presence of readily available carbon sources enhances fungal and bacterial abundance, diversity and their interplay[4,5]. However, interactions taking place at fine-scale resolution remain unclear.

Ectomycorrhizal fungi form symbioses mainly with trees in temperate and boreal forests. Numerous fungal species coexist on single host plants[6–14], such as 80–90 fungal taxa on roots of old-growth *Fagus sylvatica* trees[15]. The diversity of ectomycorrhizal fungi colonizing a plant is reflected in the morphological diversity of ectomycorrhizal root-tips, which differ in shape, color and other visual properties. Based on the assumption that the fungal tissue of one ectomycorrhizal root-tip is formed by one fungal species, morphological properties have been used, supported by microscopy and catalogued information, to identify the fungal species involved in its formation[16,17]. Studies investigating ectomycorrhizal fungi via sequencing of the internal transcribed spacer (ITS) region, however, often found more than one ectomycorrhizal fungal species present on mycorrhizal root-tips[10,12,13,18–21]. This raises the question if and how the additionally present ectomycorrhizal species may contribute to the formation of a specific morphotype and to its functionality. As fungal-specific root-tip morphologies create special microenvironments for soil organisms such as bacteria or archaea[5,22,23], the fungi building an ectomycorrhizal root-tip may influence the associated bacterial and archaeal communities.

The development of ectomycorrhizal root-tips is complex and the fungal inoculation of the root initiates the formation of the fungal mantle which can precede the association of bacteria[24]. The ectomycorrhizal mantle surface of *Fagus sylvatica* harbors high numbers of bacteria[25], and the thin interface between the host root and adhered fungal mantle is considered to represent a particular hotspot for bacterial colonization[26]. Many studies demonstrated that mycorrhizosphere-associated microbial communities can vary according to their host plants, the present soil and environmental conditions[21,25,27–31]. A rare number of studies, however, investigated fungal and/or bacterial communities at the level of individual ectomycorrhizal root-tips[18,31,32] and none has, to the best of our knowledge, explored the complex linkages between the diverse fungal and bacterial/archaeal communities associated with single root-tips. Interesting attempts have been made in this direction in the studies by Marupakula[33,34], who showed that the dominant fungal species on root-tips of *Pinus sylvestris* were associated with distinct bacterial communities[33]. Until now, however, research has mainly focused on the dominant fungal species on ectomycorrhizal root-tips, which is likely responsible for morphotype forming. As such, the influence of less abundant or even rare members of the fungal root-tip communities on bacterial associations remains unknown.

This study focused on microbial colonization patterns of ectomycorrhizal root-tips, and particularly on the question whether diverse ectomycorrhizal fungal communities influence the association of bacteria/archaea. More precisely, we hypothesized that the whole fungal community, rather than only the dominant ectomycorrhizal fungi is responsible for patterns of bacterial/archaeal communities across root-tips. To test this we used modularity, a network property that determines the presence of assemblies of bacterial/archaeal taxa that tend to inhabit the same root-tip as a given set of fungal taxa. No changes in modular structure among networks that gradually consider less abundant fungal taxa would suggest that mainly the highly abundant fungi shape the assembly of bacterial/archaeal communities on the root-tips, while gradually contrasting network structures would suggest that the consideration of rare fungi generate co-occurrence patterns, and thus also contribute to shaping the bacterial assemblies. We sequenced the 16S rRNA gene and the ITS1 region from 62 individual mycorrhizal root-tips of 13 young European Beech (*Fagus sylvatica*) trees, in addition to corresponding rhizosphere soil and root-distant soil (bulk soil). We aimed to get a comprehensive picture of the dominant and rare fungal community members on individual root-tips and to assess the associated bacterial/archaeal communities. By investigating which fungal-bacterial/archaeal taxa tend to inhabit the same mycorrhizal root-tip, we assessed tendencies in the associations of fungal and bacterial/archaeal communities. In combination with a unique conceptual framework, this study provides insights into the fungal-bacterial interplay on individual root-tips.

## Results

Screening roots of the 13 sampled trees confirmed that 80–90% of root-tips in each root system were mycorrhized (see Gorka et al.[35] for details). Mycorrhizal root-tips of sampled *Fagus sylvatica* roots showed a variety of fungal morphological appearances, differing in color, branching pattern and extensive mycelium (Fig. 1, a1–6). In a pre-experiment, we also confirmed bacterial colonization on the ectomycorrhizal fungal mantle (built from ectomycorrhizal fungal hyphae surrounding the root-tips) of *Fagus sylvatica* root-tips via scanning electron microscopy (Fig. 1, section b).

***Fagus sylvatica* ectomycorrhizal root-tips harbor specific fungal OTUs and many unique bacterial/archaeal OTUs.** Interestingly, among all investigated habitats (rhizosphere, bulk soil and mycorrhizal root-tips), mycorrhizal root-tips harbored most of the unique bacterial/archaeal OTUs (1000), representing 15% of all OTUs (Fig. 2a). In contrast, bulk soil harbored most of the unique fungal OTUs (227), accounting for 17% of all OTUs (Fig. 2b).

Analyzing the 100 most abundant shared and unique fungal and bacterial/archaeal OTUs across the three investigated habitats revealed that a large group of OTUs was solely present on mycorrhizal root-tips, while bulk and rhizosphere shared OTUs (Fig. 2c, e). Within the 100 most abundant bacterial/archaeal OTUs, OTUs almost entirely (to 90-100%) associated with mycorrhizal root-tip habitats belonged to members of bacterial orders Rhizobiales, Streptomycetes and Burkholderiales. Within the 100 most abundant fungal OTUs, OTUs with higher relative abundances on mycorrhizal root-tips than in bulk soil and rhizosphere habitats belonged to fungal orders Thelephorales, Sebacinales, Pezizales, Agaricales. (Fig. 2c, e). As expected, ectomycorrhizal and other symbiotrophic fungi tended to be associated more with mycorrhizal root-tips than with the other habitats, whereas saprotrophic and pathotrophic fungi tended to occur more in bulk soil and rhizosphere habitats (Fig. 2d). However, a large part of the fungal taxa found on root-tips was also found in the soil (Fig. 2b, d). When considering all OTUs (Supplementary Fig. 1), the patterns of OTU distribution were similar to the ones observed for the 100 most abundant OTUs. Fungal and bacterial/archaeal communities on mycorrhizal root-tips showed clear differences in their taxonomic composition compared to bulk and rhizosphere-inhabiting communities (Supplementary Figure 2). Moreover, the fungal and bacterial/

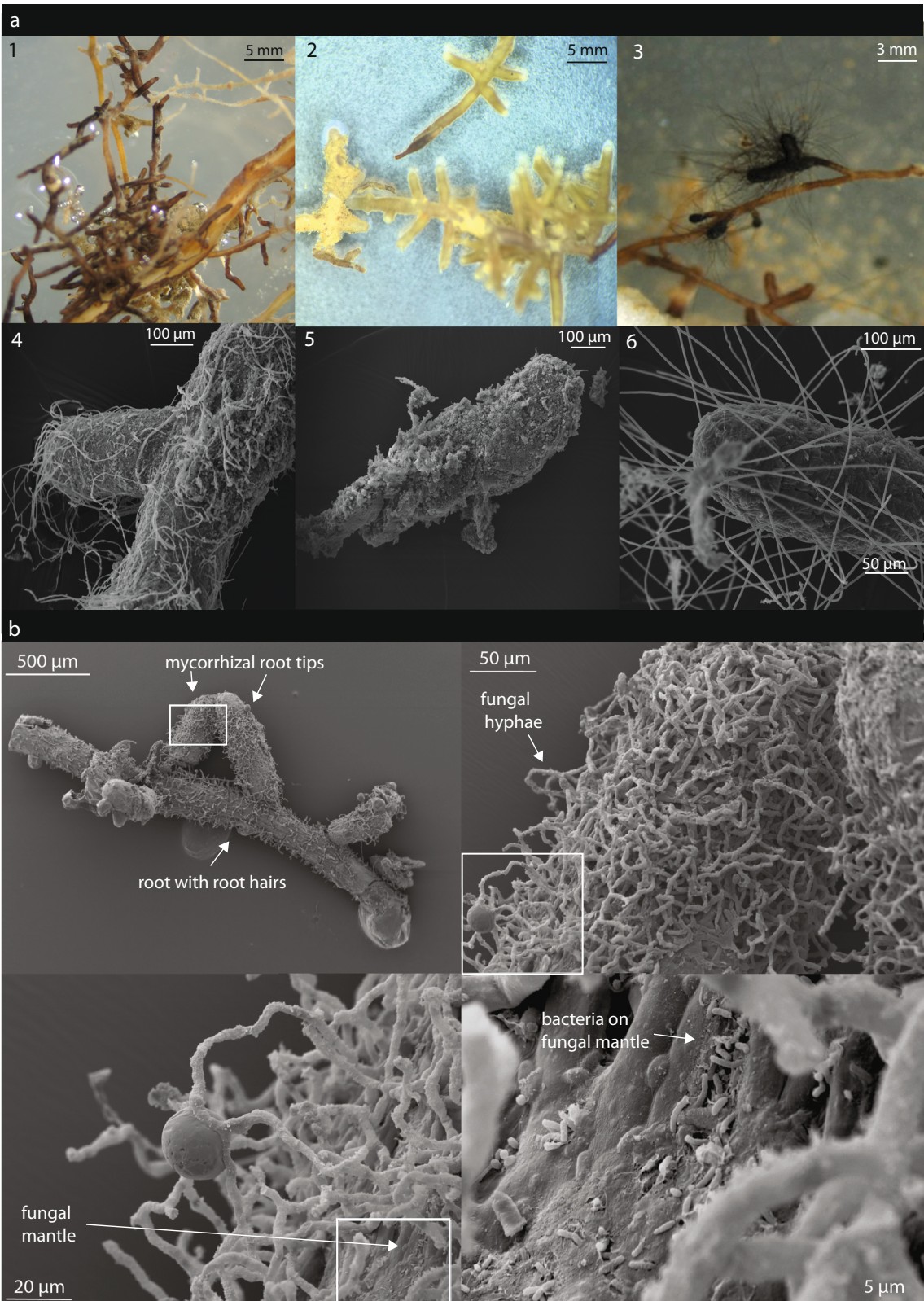

**Fig. 1 Visualizations of morphological characteristics of abundant morphotypes on mycorrhizal root-tips and their bacterial colonization.** Stereoscope images of most abundant morphotypes (**a1-3**) and corresponding scanning electron microscopy (SEM) images (**a4-6**) offer detailed insights in morphological structures. Coherent series of SEM images, starting from roots with root-tips, zoomed in on one mycorrhizal root-tip (see white boxes) that shows microbial colonization on the ectomycorrhizal fungal mantle enclosing the root-tip (**b**). Our sequencing results revealed that the root-tip depicted in **a1** and **a2** encompassed a diverse fungal community. The classification of the most abundant OTUs (>50% of the reads) revealed that **a1** belonged to *Thelephoraceae/Tomentella* and **a2** to Agaricales/*Hebelomataceae.* The root-tip depicted in **a3** was classified as *Cenococcum geophilum* based on its distinct morphology and sequence data.

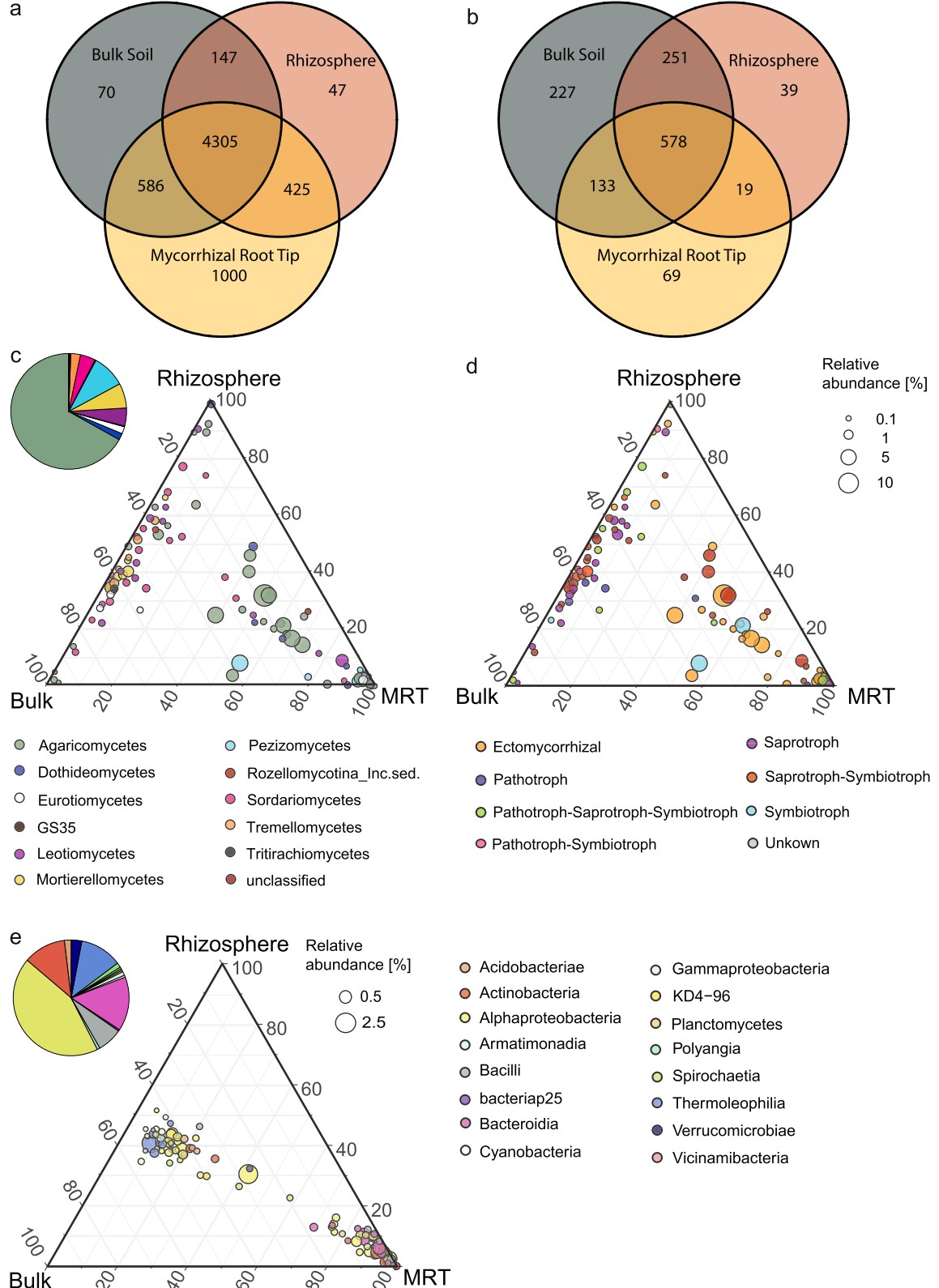

**Fig. 2 Distribution of most abundant fungal and bacterial/archaeal OTUs among bulk soil, rhizosphere and mycorrhizal root-tip habitats.** Venn diagrams depict shared and unique OTUs of sequenced bacterial/archaeal (**a**) and fungal (**b**) communities across investigated bulk soil (grey), rhizosphere (red) and mycorrhizal root-tip (yellow) microenvironments. Ternary plots depicting the distribution of 100 most abundant OTUs of fungal (**c**, **d**) and bacterial/archaeal (**e**) sequences in investigated bulk soil, rhizosphere and mycorrhizal root-tip habitats. The dot size corresponds to the average relative abundance of the OTUs across all samples. OTUs are colored by class (**c**, **e**) or fungal lifestyles (**d**). The position of the dots is determined by the occurrence and relative abundance of the OTUs in the different habitats. Points close to the corners indicate that the OTU has elevated relative abundance in this habitat compared to other habitats. Pie charts depict the taxonomic composition of the 100 most abundant OTUs in all habitats.

archaeal community compositions on mycorrhizal root-tips were significantly less diverse (Shannon diversity index; fungal dataset: ANOVA, Tukey HSD, $p < 0.001$; bacteria/archaea: Dunn test, Kruskal-Wallis multiple comparison, $p < 0.001$; Supplementary Fig. 3) and significantly different from bulk and rhizosphere soil (Bray-Curtis dissimilarity; PERMANOVA, $p = 1e^{-04}$; Supplementary Fig. 3). However, dispersion between groups was not homogenous (ANOVA, $F = 21.99$, $p > 0.001$), most likely due to the high group variance dispersion in mycorrhizal root-tip samples (Permdist, $p = 1e^{-04}$; Supplementary Fig. 4).

**Mycorrhizal root-tips were characterized by one to three dominating fungal OTUs and a more even bacterial/archaeal distribution.** Most root-tips fungal communities were dominated

by a few fungal OTUs, which constituted more than 50% of the relative read abundance. Most mycorrhizal root-tips harbored one to three dominant OTUs with relative abundance ranging from 45% up to 97% and many rare fungi (relative read abundance <5%, on average 904 OTUs per root-tip) (Fig. 3). OTUs that dominated the community of individual root-tips were mostly associated with the fungal orders Thelephorales, Agaricales, Pezizales and Sebacinales (Fig. 3). The dendrogram demonstrates that the most similar fungal communities were derived from root-tips stemming from different plants.

We identified 19 highly abundant fungal OTUs (relative abundance >50% on individual root-tips), which likely contribute to the formation of the mantle tissue and the morphological structure of each individual root-tip (labeled OTUs in Supplementary Fig. 5). They consisted mainly of ectomycorrhizal

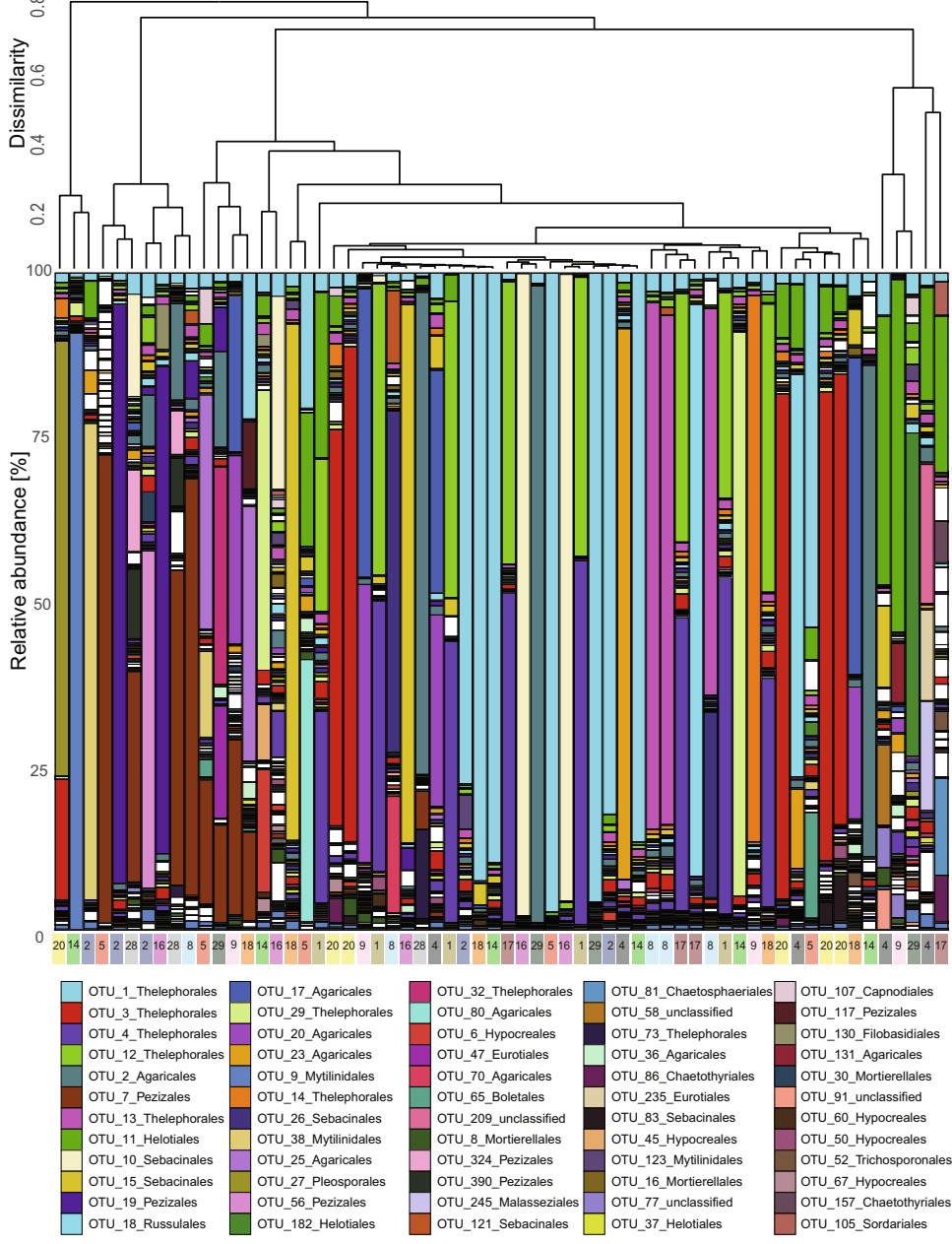

**Fig. 3 Taxonomic assignment of fungal OTUs on individual mycorrhizal root-tips.** Relative read abundance (%) of fungal OTUs associated to the investigated 62 mycorrhizal root-tips samples. The 60 most abundant fungal OTUs are depicted in color; all other (less abundant) OTUs are depicted in white. The dendrogram indicates clustering of root-tips based on fungal community composition (Bray-Curtis dissimilarity). Each bar represents one mycorrhizal root-tip sample, the number underneath refers to the tree from which the root-tip originated. Same trees are highlighted in the same color.

members of Thelephorales, Pezizales, Sebacinales and Agaricales (see Supplementary Table 2 for detailed taxonomic classification). Seven of these OTUs (OTU_1, 2, 4, 7, 11, 15, 23), encompassing members of the genera *Inocybe*, *Tomentella*, *Peziza* and *Laccaria*, were prevalent in more than 90% of all root-tips and thus part of the shared fungal community among all investigated root-tips (see labeled OTUs in Supplementary Fig. 5).

The bacterial/archaeal community composition was more even among the investigated root-tips (Supplementary Fig. 2), containing less dominant OTUs (relative average abundance of 8% across all root-tips) that were mainly members of Proteobacteria and Actinobacteria. Reads assigned to Proteobacteria represented more than 50% on 35 root-tips, specifically Alpha- and Gammaproteobacteria, while reads assigned to Actinobacteria were predominant on 23 root-tips and represented more than 50% of the reads in 16 samples. Reads assigned to Acidomicrobia, Verrucomicrobiae, Bacteroidia, Chloroflexia, Cyanobacteria, Planctomycetes, Firmicutes and Gemmatimonadetes were less abundant, but present on the majority of the sequenced mycorrhizal root-tips (Supplementary Fig. 2). When analyzing all possible combinations of shared fungal or bacterial/archaeal OTUs across the investigated mycorrhizal root-tip samples, we were able to identify four fungal OTUs (belonging to fungal orders Thelephorales and Agaricales) and 15 bacterial OTUs (belonging to bacterial orders Bacillales, Burkholderiales, Chthoniobacterales, Corynebacteriales, Frankiales and Rhizobiales) that co-occurred on all investigated mycorrhizal root-tips (Supplementary Fig. 6, see Supplementary Table 3 for detailed taxonomic classification of co-existing fungal and bacterial OTUs).

### Bacterial/archaeal communities on ectomycorrhizal root-tips are shaped by the entire fungal community including less abundant fungi.

The non-random community structure on ectomycorrhizal root-tips indicates a link between the distribution of associated fungi and bacteria/archaea. This was shown by the significant difference of the observed modularity in most bipartite networks when compared to their respective null models (Table 1). In addition, we saw a gradual increase of significance in modularity when comparing network "10%" (restricted network including only fungal OTUs with >10% relative abundance in individual samples) to network "0.1" (most complete network including fungal OTUs with relative abundance >0.1% in individual samples; Tables 1 and 2). The individual networks encompassed various numbers of modules, each of which consisted of different fungal and bacterial OTUs (Table 2).

Network "10%" was significantly modular ($z = 56.52$) compared to 100 randomized networks serving as a null model and consisted of four modules comprising 40 fungal OTUs (see Supplementary Table 4 for detailed taxonomic classification). Similarly, network "5%" was highly modular ($z = 95.47$) and was divided into six modules encompassing 54 fungal OTUs (see Supplementary Table 5 for detailed taxonomic classification). The significance of modularity peaked in network "0.1%" ($z = 851.24$), which included less abundant fungal OTUs with >0.1% relative abundance (in total 416 fungal OTUs; Tables 1, 2). Within all three networks, each module differed in its fungal and bacterial composition (Fig. 4). While some modules were dominated by ectomycorrhizal members of Agaricales, Sebacinales, Pezizales, Russulales, Thelephorales, other modules showed a more diverse distribution of various members of ectomycorrhizal and non-ectomycorrhizal fungal guilds (Fig. 4). Bacterial community composition was dominated by members of Proteobacteria, Acidobacteria, Actinobacteria, Bacteroidota and Planctomycetota, but their relative abundances changed across the different modules of each network (Fig. 4).

To assess whether the whole fungal community, and not only the most abundant ectomycorrhizal fungi, shape the bacterial/archaeal communities present on each root-tip, we evaluated the consistency of modules' composition across networks with different fungal OTU abundance cut-offs. We compared the bacterial/archaeal and fungal taxa assignments into modules between network "10%" and network "0.1%", as well as network "5%" and network "0.1%" (Fig. 5).

Including rare fungi altered the structure of the network allowing the emergence of specific bacterial/archaeal associations with rare fungi. We observed two patterns in the structure of the networks that explain the effect of rare fungi on the fungal-

---

**Table 1 Modularity parameters of all five calculated networks with different fungal OTU relative abundance cut-offs. B: Bacteria/Archaea; F: Fungi.**

| Network | fungal OTU cut-off [% relative abundance] | Total OTUs after fungal cut-off[a] | z-Score | Modules |
|---|---|---|---|---|
| "25%" | 25 | 26 F, 6632 B | −3.55 | 5 |
| "10%" | 10 | 40 F, 6632 B | 56.52 | 4 |
| "5%" | 5 | 54 F, 6632 B | 95.47 | 6 |
| "1%" | 1 | 126 F, 6632 B | 258.72 | 5 |
| "0.1%" | 0.1 | 416 F, 6632 B | 851.24 | 4 |

[a]Please note that the numbers of bacterial OTUs were the same across all networks because we only applied a cut-off to fungal OTUs.

---

**Table 2 Number of fungal and bacterial/archaeal OTUs in each module of the networks.**

| | Networks | | | | | | | | | |
|---|---|---|---|---|---|---|---|---|---|---|
| | 25% | | 10% | | 5% | | 1% | | 0.1% | |
| | F | B | F | B | F | B | F | B | F | B |
| Modules | 11 | 1835 | 10 | 2023 | 2 | 675 | 3 | 165 | 6 | 136 |
| | 4 | 1446 | 6 | 1091 | 13 | 1721 | 44 | 2287 | 268 | 2868 |
| | 2 | 469 | 4 | 1268 | 5 | 898 | 52 | 2389 | 126 | 3380 |
| | 5 | 1533 | 20 | 2250 | 17 | 1765 | 4 | 283 | 16 | 248 |
| | 4 | 1339 | | | 9 | 953 | 23 | 1508 | | |
| | | | | | 8 | 620 | | | | |
| Sum of OTUs | 26 | 6622[a] | 40 | 6632 | 54 | 6632 | 126 | 6632 | 416 | 6632 |
| Number of modules | 5 | | 4 | | 6 | | 5 | | 4 | |

Each network (in columns) has a different number of modules (depicted in rows) and each module consists of a certain number of fungal (F) and bacterial/archaeal (B) OTUs.
[a]Please note that when considering the relative abundance cut-off of 25%, only 6622 bacterial/archaeal OTUs were used in the modularity analysis. This is due to the fact that some bacterial OTUs were not interacting with the fungal OTUs remaining when considering the 25% cut-off.

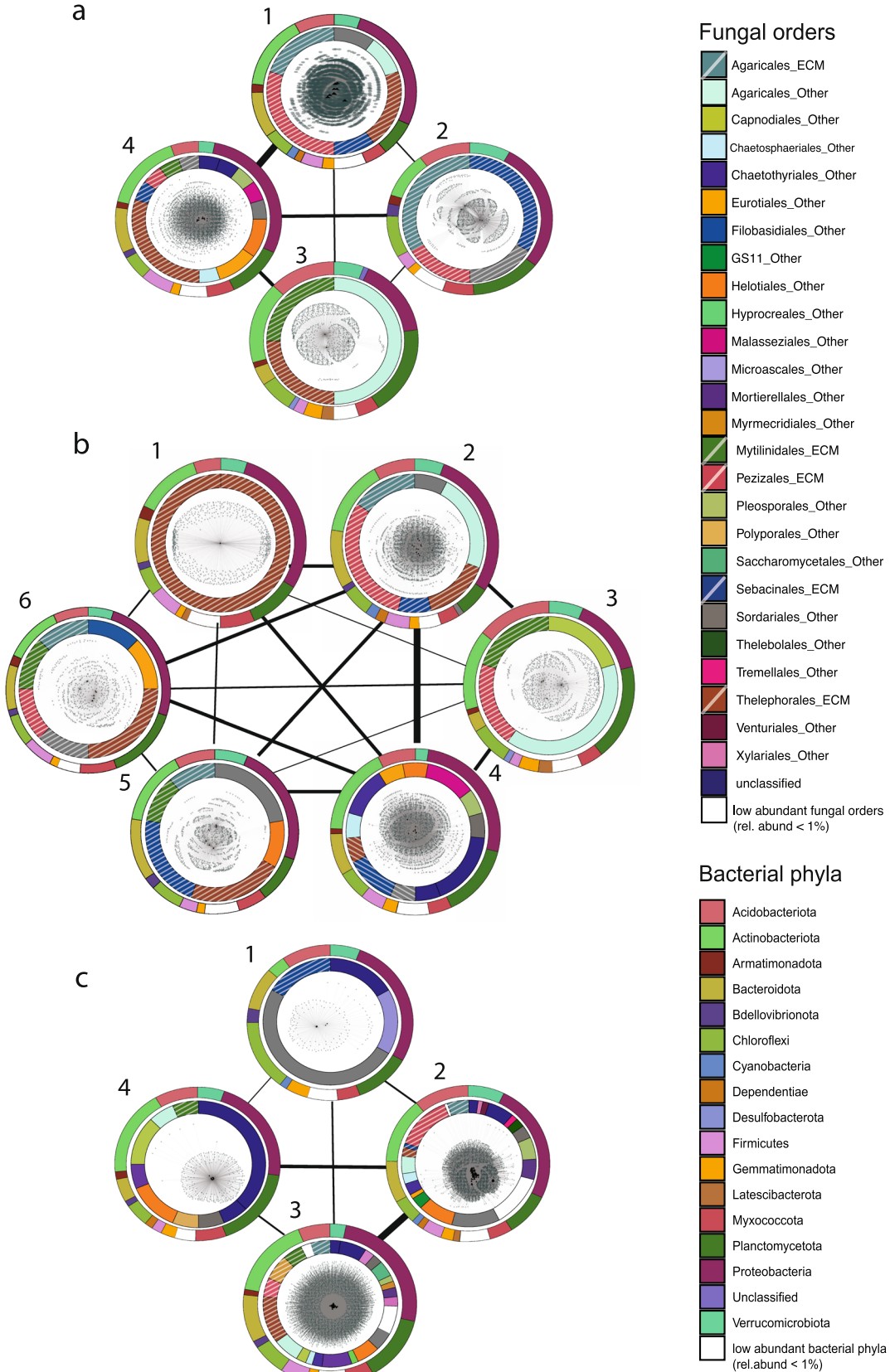

**Fungal orders**

- Agaricales_ECM
- Agaricales_Other
- Capnodiales_Other
- Chaetosphaeriales_Other
- Chaetothyriales_Other
- Eurotiales_Other
- Filobasidiales_Other
- GS11_Other
- Helotiales_Other
- Hyprocreales_Other
- Malasseziales_Other
- Microascales_Other
- Mortierellales_Other
- Myrmecridiales_Other
- Mytilinidales_ECM
- Pezizales_ECM
- Pleosporales_Other
- Polyporales_Other
- Saccharomycetales_Other
- Sebacinales_ECM
- Sordariales_Other
- Thelebolales_Other
- Tremellales_Other
- Thelephorales_ECM
- Venturiales_Other
- Xylariales_Other
- unclassified
- low abundant fungal orders (rel. abund < 1%)

**Bacterial phyla**

- Acidobacteriota
- Actinobacteriota
- Armatimonadota
- Bacteroidota
- Bdellovibrionota
- Chloroflexi
- Cyanobacteria
- Dependentiae
- Desulfobacterota
- Firmicutes
- Gemmatimonadota
- Latescibacterota
- Myxococcota
- Planctomycetota
- Proteobacteria
- Unclassified
- Verrucomicrobiota
- low abundant bacterial phyla (rel.abund < 1%)

**Fig. 4 Structure of the modules in the investigated networks.** Ring graphs visualizing (**a**) network "10%", (**b**) network "5%" and (**c**) network "0.1%". Each network shows the structure of its modules (modules are numbered) and the surrounding rings correspond to the taxonomic composition of the respective module. The inner ring represents the composition of fungal orders in each module, ectomycorrhizal lifestyles are depicted striped. The outer ring represents the distribution of bacterial phyla. All taxa with a relative abundance <1% are condensed and depicted in white. The strength of the connection between the modules of each network corresponds to the thickness of the connecting black line.

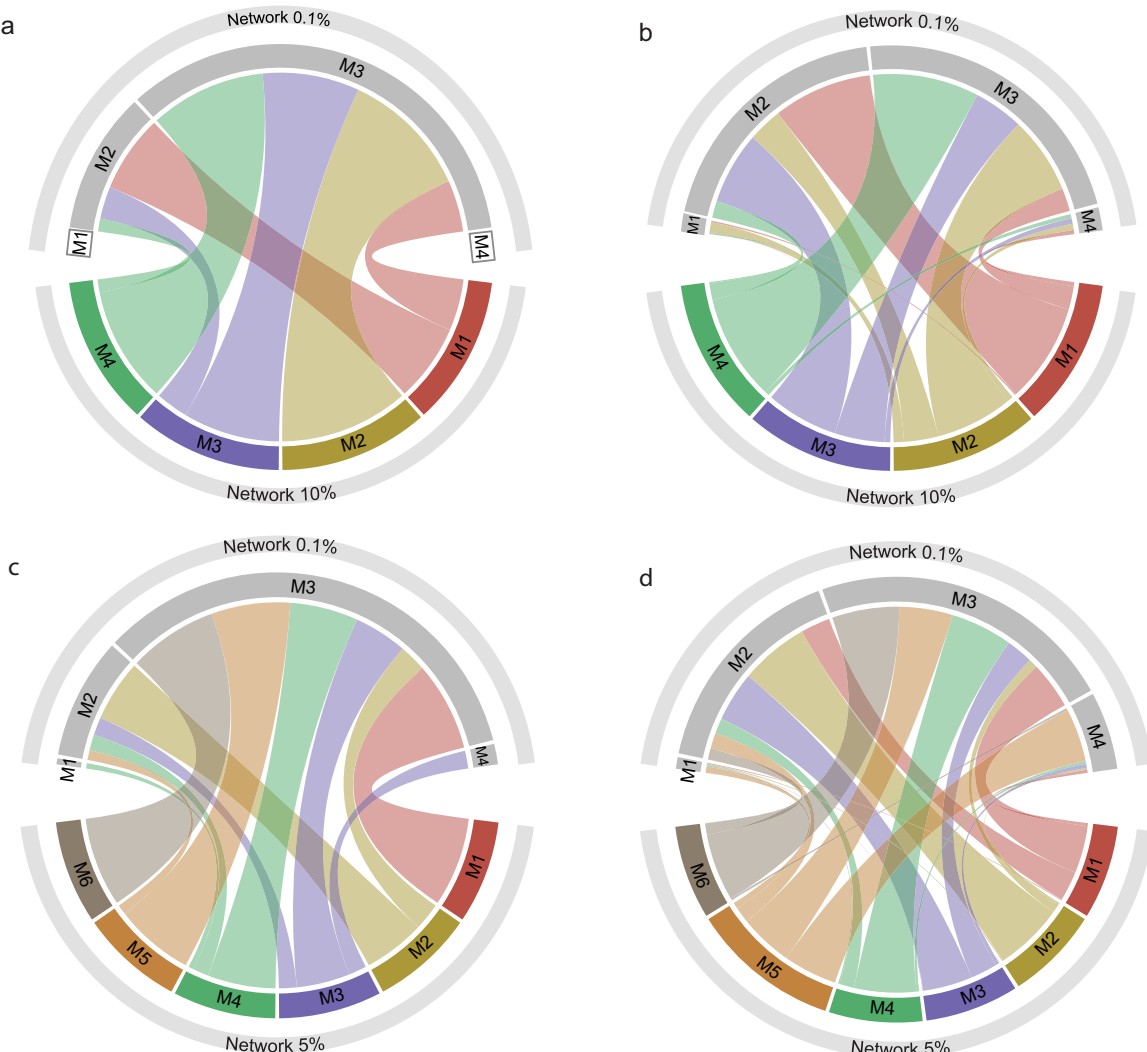

**Fig. 5 Chord diagrams comparing the consistency of modules' composition across networks with different fungal OTU abundance cut-offs.** The fungal (**a**, **c**) and bacterial/archaeal taxa (**b**, **d**) assignments into modules of different networks are shown. Modules of network "0.1%" (encompassing fungal OTUs >0.1% relative abundance) are depicted in the upper part of each chord diagram and were compared to modules of network "10%" (encompassing fungal OTUs >10% rel. abundance, panel **a** and **b**) and network "5%" (encompassing fungal OTUs >5% rel. abundance, panel **c** and **d**). Only those fungi present in both, network "0.1%" and "5%" or "10%" (i.e. more abundant fungi) are compared. All modules are presented, but those composed exclusively of rare fungi (present only in network "0.1%") appear empty (i.e. M1 and M4 in **a**).

bacterial/archaeal assemblies. We show statistical evidence for those patterns that relate to two non-exclusive ecological processes. First, comparing the module patterns across networks showed that network "0.1%" (comprising also rare fungi >0.1% relative abundance) contained two modules (M1, M4) solely composed of rare fungal taxa (with a relative abundance of less than 5% and more than 0.1%), which were not included in the restricted network "10%" or "5%" (consisting of only abundant fungal OTUs with relative abundance >10% and >5%, respectively; Fig. 5). This pattern was clearly visible in the comparison with network "10%" (Fig. 5a) and less strong, but still distinct in the comparison with network "5%" (Fig. 5c, M1 and M4 of network "0.1%" contain only a few abundant fungi and mostly rare fungi). Second, including the rare fungi also changed which abundant fungi tended to be associated to the same bacteria/archaea. Comparing networks revealed that highly abundant fungal taxa change their module's partners when also considering rare fungi; taxa belonging to the same module in the restricted networks did not share the same module in the most complete network (network "0.1%"). The proportion of highly abundant

fungal taxa of networks "10%" and "5%" (>10% and >5% relative abundance) grouping together within the same module in the compared network "0.1%" was significantly smaller than 1 for both comparisons, network "10%" and "0.1%" (mean = 0.28, t = −7.99, df = 6, p value < 0.001) and network "5%" and "0.1%" (network "5%"; mean = 0.20, t = −13.97, df = 13, p value < 0.001; a proportion of 1 would indicate that 100% of the fungal taxa belonging to the same module in one network also share a module in the other network).

The bacterial/archaeal taxa were distributed across all modules of both network pairs, although a bigger proportion of these taxa was shared with M2 and M3 of network "5%" and "10%" (Fig. 5b, d) in comparison with network "0.1%".

## Discussion

In this study we showed that the unique microenvironment of established *Fagus sylvatica* ectomycorrhizal root-tips harbors specific microbial communities with particular co-occurrence patterns. Our data suggest that mycorrhizal root-tips host diverse fungal communities as well as unique bacteria/archaea. These

specific bacterial/archaeal communities associated to mycorrhizal root-tips are selected by the fungal community present on each root-tip. Furthermore, our results demonstrate that not only the most abundant fungi, but also less abundant fungi are associated with the bacterial/archaeal community within a root-tip and alter the structure of the bipartite network.

Each mycorrhizal root-tip was inhabited by one to three dominant ectomycorrhizal OTUs, which comprised more than 50% of the reads. It is likely, that the fungal mantle tissue of root-tips is mostly formed by those highly abundant fungi, which subsequently shape the morphological structure of each individual root-tip. As such, they constitute preferable microhabitats for bacterial/archaeal, as well as (lower abundant) fungal colonization.

We observed vastly differing fungal communities amongst single root-tips of the same plant, which can be explained by competition, including priority effects, facilitation or parasitism among species[10,36–38]. In particular, positive interactions and competition between ectomycorrhizal fungal species determine root colonization[39]. Especially multi-host ectomycorrhizal fungi colonizing different host trees, such as members of the genera *Inocybe*, *Cenoccocum*, *Laccaria* and *Russulales*, are known to be strong competitors[17,36]. They were also among the most prevalent fungi in our system, which confirms their consistent presence across the investigated root-tips.

Based on scanning electron microscopy (Fig. 1, section b), we detected dense bacterial colonization on the ectomycorrhizal mantle surface of *Fagus sylvatica*, which is in accordance with previous research[25]. Mycorrhizal fungi themselves exude carbon compounds and their varying quality and quantity influences mycorrhizosphere bacterial/archaeal community composition[33,40–43]. In addition, root exudates can be altered by fungi, leading to quantitative and qualitative changes in root exudates between rhizosphere and mycorrhizosphere, selectively shaping the associated microbial communities, the so called mycorrhizosphere effect[4,27,44]. The significant difference in community richness, evenness and diversity between mycorrhizal root-tip and soil-related bacterial/archaeal communities indicates a strong mycorrhizosphere effect in our system[4]. However, since the dispersion among group was not homogenous, the results must be interpreted with care.

Dominant bacteria on the root-tips, such as members of Burkholderiales, Streptomycetales and Rhizobiales were previously found in ectomycorrhizal systems: Burkholderiales, the most abundant colonizers of soil fungi[45], have been detected on mycorrhizal root-tips[29,33,46,47] and endophytes of *Fagus sylvatica*[25]. Members of Rhizobiales have been found on ectomycorrhizal root-tips of *Pinus muricata*[46], and peridia of *E. granulatus* and truffles have been found to provide a suitable niche for *Bradyrhizobia*[48–50]. Many bacterial strains have been reported to enhance ectomycorrhizal formation[45,51,52] and improve plant nutrition and health. Members of Streptomycetales, for instance, are known as important symbiosis modulators by inhibiting growth of pathogenic fungi and promoting ectomycorrhizal growth formation[53]. In crop plant systems, such as soybean roots, rice roots or sugarcane rhizosphere, it has been reported that Burkholderiales, Streptomycetales and Rhizobiales act as plant growth promoting bacteria and are the core-responsive bacteria under drought conditions, mediating drought tolerance[54–56]. However there is still a lack of knowledge about the role of non-easily culturable bacteria living in the mycorrhizosphere.

The large group of unique bacterial/archaeal OTUs present on mycorrhizal root-tips indicates that ectomycorrhizal fungi create preferred microhabitats for distinct bacteria/archaea[26]. The non-random community structure of the investigated mycorrhizal root-tips suggests that they are determined by the fungal taxa present. Fungal hyphae can select for certain bacteria and determine their community composition[57,58] and vice-versa, mycorrhizosphere bacteria can act in a fungi-specific manner and stimulate mycorrhizal formation in combination with certain fungi, but inhibit the symbiosis with others[53,59].

The colonization of mycorrhizal root-tips is complex, triggered by a multitude of signaling pathways in the symbiotic interfaces of host and fungi, preceding fungal inoculation of the roots[24,60]. Bacterial colonization follows the formation of the initial fungal mantle. By analyzing the fungal patterns in module assignments among the different networks, we identified potential underlying processes contributing to shifts in the structure of co-occurring bacterial/archaeal communities due to fungal-bacterial/archaeal interactions. We showed that not only highly abundant, but also rare fungal OTUs shape the fungal-bacterial/archaeal interactions, highlighting the importance of comprehensive analysis of fungal communities on individual root-tips, that is not based solely on the characterization of dominant fungi. However, we do not claim to have covered the entire rare fungal community associated to mycorrhizal root-tips, as this is dependent on factors such as the used extraction method and sequencing depth. Based on the distribution of abundant and rare fungi among the modules (Fig. 5), we propose a framework (Fig. 6) to assess two non-exclusive processes contributing to shifts in the structure of the resulting network when all fungi (including the rare ones) are considered compared to only the abundant ones. The consideration of rare fungi can result in (a) the emergence of specific bacterial/archaeal associations to the rare fungi, and (b) changes in the similarity patterns of bacterial/archaeal associations between the highly abundant fungi. On the one hand, if changes are induced by the emergence of specific bacterial/archaeal associations to rare fungi, we expect these fungi to be grouped into new modules in the most complete network (including all fungi) (Fig. 6, panels 1.1 or 1.2). On the other hand, if changes are induced by a split of the group of bacteria/archaea that were previously associated with the same highly abundant fungi, we expect that abundant fungi from the same module in the more restricted network would not be grouped together in the most complete network (Fig. 6, panels 1.1 or 2.1). These two processes can affect the shifts in the network either independently or simultaneously. If neither of those processes occur, shifts between the networks will be minimized. No changes between the networks structure will occur when rare fungi are evenly distributed across modules in the most complete network and highly abundant fungi keep sharing the same module (Fig. 6, panel 2.2).

Based on our network analysis we saw significant patterns in both outlined processes (Fig. 6, panel 1.1). Including rare members of the fungal community resulted in newly emerging modules, exclusively containing rare fungi. Furthermore, highly abundant fungi did not share the same module when rare fungi were considered (Fig. 5).

The generation of new modules harboring only rare fungi (Fig. 5a, c) suggests that these low abundant fungi could facilitate novel bacterial/archaeal assemblages. In addition, the uneven distribution of highly abundant fungi among the different modules of the most complete network suggests that the association patterns between bacteria and some highly abundant fungi are weak, and they might tend to similarly associate to other highly abundant fungi when more detailed information about their association patterns is included in the network by considering the presence of rare fungi.

In summary, our network analysis demonstrates that rare fungi can alter the potential for microbe-microbe interactions, leading to new fungal-bacterial community assemblies. Our study further emphasizes the importance of focusing on small-scale fungal-bacterial/archaeal co-occurrences among individual mycorrhizal root-tips, as it holds the potential to deepen our understanding of

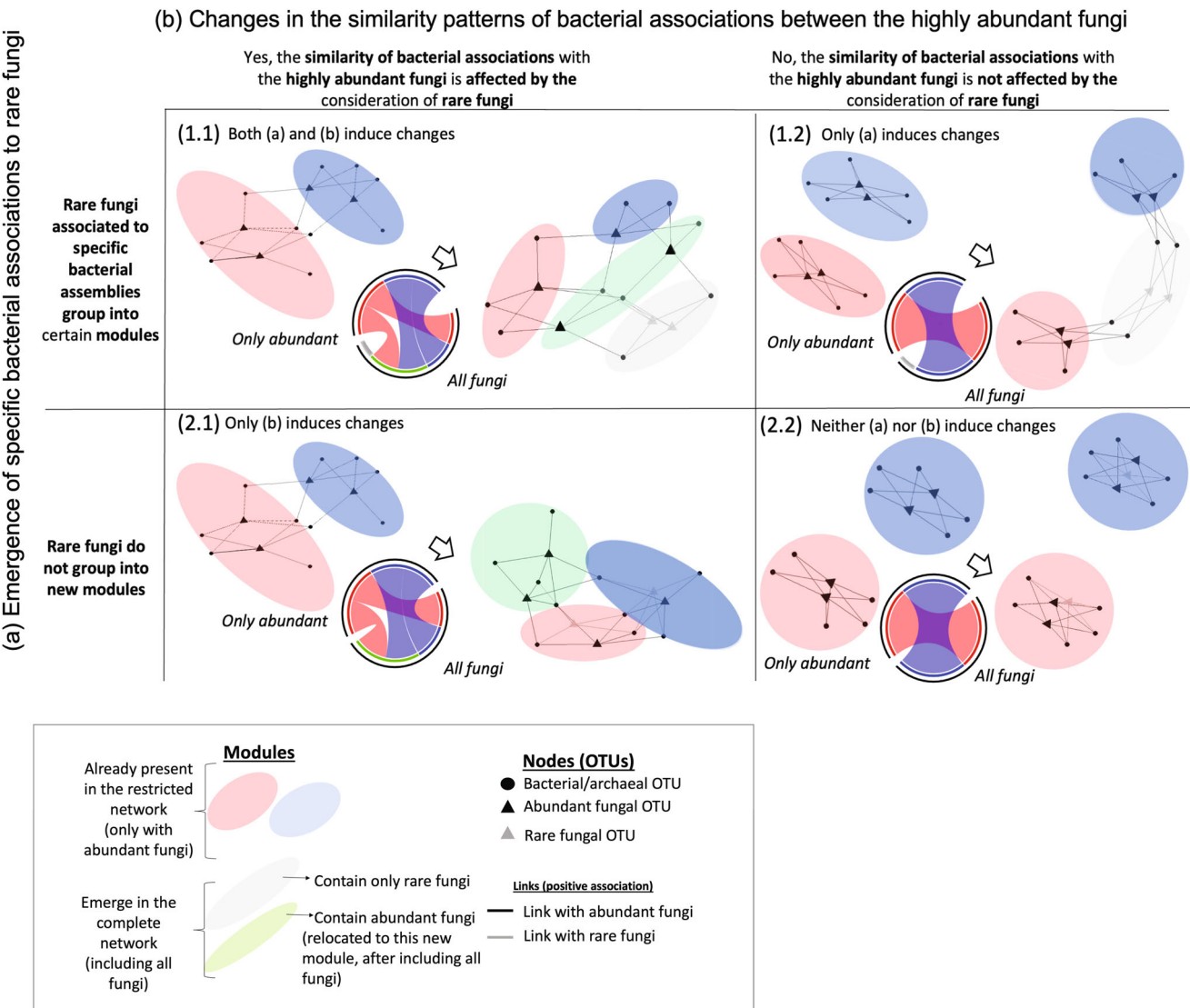

**Fig. 6 Conceptual framework on processes shaping the effect of fungi on the bacterial/archaeal communities associated with ectomycorrhizal root-tips.** Morphological features of ectomycorrhizal root-tips provide distinct microenvironments for smaller microorganisms such as bacteria and archaea. Such morphological features, which select for certain bacterial/archeal communities, can be shaped by only the most abundant fungi, but may also be influenced by the entire fungal community, including the less abundant ones. Depending on the influence of rare fungal taxa, fungal-bacterial/archaeal bipartite networks may show different structures if only highly abundant fungi or the whole fungal community are considered. Here we present a conceptual framework for interpreting such emerging differences in network structure: We assume that a shift in network structure after considering rare fungi can result from two non-exclusive processes: a) the emergence of specific bacterial associations to the rare fungi, and b) changes in the similarity patterns of bacterial associations with the highly abundant fungi. If changes are induced by the emergence of specific bacterial associations to rare fungi, these fungi will tend to be grouped into certain (new) modules in the network that considers all fungi (1.1 and 1.2, notice the grey module only present in the most complete network). If changes are induced because rare fungi alter the relationship between bacteria/archea and abundant fungi (f.e. by a split of the group of bacteria/archaea that were previously associated to the same highly abundant fungi) we expect to see that fungi which shared the same module in the 'abundant fungi only' network will be in separate modules in the 'all fungi' network (1.1 and 2.1, notice the green module consisting of a mix of colors (red and blue) in the lower part of the chord diagrams). These two processes can also simultanously affect the shifts in the network (1.1), and if none of them take place the shifts will be minimized (2.2). Networks for 'only abundant' and 'all fungi' are represented in the example chord diagrams in the middle of each panel (upper half shows only abundant fungi, lower half shows all fungi, colors depict different modules, as in Fig. 5), which visualize a potential difference in module structure. The elipses are colored according to the colors in the chord diagram: blue and red refers to non-changing modules in networks; grey refers to new modules that emerge due to the consideration of rare fungi and only contain rare fungi; green refers to modules that emerge due to changes in the highly-abundant fungi. Black triangles: abundant fungi, grey triangles: rare fungi, black circles: bacteria/archaea.

the complex interplay of the symbiotic partners in the widespread and vital tripartite ectomycorrhizal symbiosis.

## Methods

Approximately three- to four-year-old *Fagus sylvatica* trees were sampled from a beech forest in Klausen-Leopoldsdorf (Austria). Based on stereoscopic examination

of selected plant roots, all trees showed abundant ectomycorrhization at the time of replanting. Specifically, twenty-seven trees were transferred into individual split-root boxes (as this study was part of a bigger experiment, see detailed information on experimental setup in previous studies using the same plants[35,61]). The trees were planted in a mixture of soil (collected from the tree sampling site, A-horizon, 4 mm sieved) and perlite (soil:perlite (8:1, v/v)[35,61] and grown in a greenhouse under ambient (outdoor) sunlight and temperature for a year. After 12 months of

plant growth, 13 plants were sampled for this study (Supplementary Table 1). During the plant harvest, boxes were carefully taken apart. Plant root systems were gently shaken, soil falling off, together with the soil that remained in the box, was 2 mm sieved and declared bulk soil. Root pieces were cut off and shaken in 1× Phosphate-buffered saline (PBS, pH = 8) for 30 min. After root pieces were transferred into new tubes, the remaining soil slurry was centrifuged (8000× g, 4 min), the supernatant discarded, and the pellet considered rhizosphere soil.

To obtain mycorrhizal root-tips (about 1-3 mm long terminal ends of the individual roots), roots were washed in tap water, the root system split into at least 6 parts and screened for total mycorrhization using stereoscopes[35,61] (see an example for one of the root parts in Supplementary Material Fig. S2 in Gorka et al.[35]). Mycorrhization (i. e. the percentage of plant root-tips colonized by ectomycorrhizal fungi, estimates on average 85%) was similar across all investigated root systems. Individual mycorrhizal root-tips (for an example, see Supporting Information Fig. S1 in Mayerhofer et al.[61]) were sampled from different root parts. All samples were frozen at −80 °C until further processing.

**Total nucleic acid extraction**. Total nucleic acids were extracted according to Angel et al.[62], using a mechanical and chemical lysing approach (as proposed in[63] by combining bead-beating with phenol-chloroform extraction. 0.4 g of bulk soil and rhizosphere soil were used. For mycorrhizal root-tips (directly weighed into bead-beating tubes), the extraction protocol was slightly adapted by performing four bead-beating steps and by subsequently transferring only the aqueous phase without the phenolic phase. Bulk and rhizosphere samples were resuspended in 200 μl Low TE Buffer and mycorrhizal root-tips in 30 μl. Immediately after extraction, humic substances were removed with the OneStep[TM]PCR Inhibitor Removal Kit (Zymo Research, Irvine, CA, USA). DNA was quantified by using the Quant-iT[TM] PicoGreen®dsDNA assay (ThermoFisher Scientific, Waltham, MA, USA).

**Sample preparation for identification of microbial and fungal communities via MiSeq Illumina sequencing**. A multiplexed barcoded amplicon sequencing approach[64] was used to amplify the 16S rRNA gene and the fungal ITS1 region from extracted DNA samples. Briefly, the 16S rRNA was amplified with the primer set H-515F- mod and H-806R-mod targeting general bacteria and archaea (515 F: 5'-H- GTGYCAGCMGCCGCGGTAA-3'; 806 R: 5'-H-GGACTACNVGGGTWT CTAAT- 3'). For details on amplification protocols see supplementary materials (Supplementary Methods 1). The fungal ITS1 region was amplified with primers ITS1F (5'-H- GCTATGCGCGAGCTGCCTTGGTCATTTAGAGGAAGTAA -3') and ITS2 (5'-H-GCTATGCGCGAGCTGCGCTGCGTTCTTCATCGATGC-3') as described in detail in Gorka et al.[35] and sequencing was performed using the Illumina MiSeq platform, v3 chemistry, and 2 × 300 bp sequencing mode at Microsynth AG (Balgach, Switzerland).

**Visualization techniques of mycorrhizal root-tips and associated microorganisms**. Mycorrhizal root-tips were fixed in 1% glutaraldehyd (GA) for 2 h and washed by exchanging GA with $K_2HPO_4$:$KH_2PO_4$ buffer three times. To preserve the native structure of the samples, they were treated with 1% osmiumtetroxide ($OsO_4$) solution for 2 h and washed with MQ water thrice. Samples were dehydrated through an ethanol/acetone series (50%-100%) and subsequently critical point dried (25 cycles, EM CPD300, Leica Microsystem, Wetzlar, Germany) by replacing water with $CO_2$ through the intermediate fluid acetone. Critical point dried samples were fixed on stubs with carbon stickers and sputter-coated with palladium-platinum (Pd/Pt) for 100 secs at 80 mA (High Resolution Fine Coater JFC-2300 HR, JEOL, Freising, Germany). Morphological structures of different mycorrhizal root-tips and their colonization with microorganisms were investigated by scanning electron microscopy (SEM Jeol IT 300, JEOL, Freising, Germany).

**Data analyses**. Paired sequence reads were merged using BBmerge v.37.61[65] with "strict" setting (requiring exact match), a minimum overlap of 50 bp after clipping 3'-prime ends with quality scores below 20. In total, 884136 16S rRNA gene and 498541 ITS reads were generated. Exact amplicon sequence variants (ASVs) were determined with DADA2[66] based on the entire dataset ("pool=T" setting) with standard settings. The ASVs were further grouped into percentage-identity-independent "operational taxonomic units (OTUs)" with SWARM2[67] in fastidious mode with limit of a large swarm for grafting set at 20. Taxonomy was assigned to OTU-centroids by Last-common-ancestor (LCA) algorithm using rRNA-secondary-structure-aware SINA aligner v.1.2.11[68] and the SILVA SSU138 database[69]. Unite database (version 8.2) was used for fungal taxonomic classification. Fungal lifestyles were defined based on genus level or on higher levels in case genus was not available, according to the FUNguild database implemented in R[70].

Highly abundant OTUs from the extraction blanks were removed from the 16S rRNA dataset (11 OTUs) and from the ITS1 dataset (7 OTUs). Further, OTUs not classified at kingdom or domain level were excluded from both datasets. OTUs classified as mitochondria or chloroplasts (58 OTUs) were removed from the 16S rRNA dataset and OTUs not classified as fungi at kingdom level (44 OTUs) were removed from the fungal ITS1 dataset. The average number of reads per individual

sample after removing reads that correspond to these aforementioned OTUs was ~6000 for ITS and ~9000 for 16S rRNA genes (detailed information on the final number of reads per sample can be found in the Supplementary Data 1). The final datasets contained 1323 fungal OTUs (encompassing 13 phyla, 41 classes) and 6784 bacterial OTUs/ 23 archaeal OTUs (encompassing 42 phyla, 107 classes, 309 families). Because archaeal OTUs represented only 0.3% (23 OTUs) of all 16S OTUs, bacterial and archaeal OTUs were analyzed together.

**Statistics and reproducibility**. For comparisons among bulk soil, rhizosphere and mycorrhizal root-tips, samples derived from seven trees were used (Supplementary Table 1), resulting in eight bulk samples, seven rhizosphere samples and 31 mycorrhizal root-tip samples. For all analyses comparing mycorrhizal root-tips amongst each other, 31 mycorrhizal root-tip samples of six additional trees (4-6 root-tips per tree) were included in the analysis, leading to a total of 62 mycorrhizal root-tip samples derived from a total of 13 trees (Supplementary Table 1).

Data analyses and statistics were performed in R (version 3.6.3). Packages "phyloseq"[71] and "vegan"[72] were used for sequence data analyses. Graphs were plotted in "ggplot2"[73]. Package "ggtern"[74] was used for ternary plots and package "UpSetR"[75] was used to create upset plots. See supplementary material (Supplementary Methods 2) for details on statistics.

**Network analysis**. Most network analyses of soil microbial communities rely on a statistical inference of association metrics based on pairwise correlations of taxa across a sufficiently large number of soil samples (i.e. co-occurrence networks)[76]. This is necessary because it is impossible to directly quantify small-scale microbial interactions, in the relatively large size of soil samples. In our approach we circumvented this need by performing a sampling design at an ecologically meaningful spatial scale (as described for instance in Fortuna et al.[77] and Montesinos-Navarro et al.[78]), that allows to observe which fungi and bacteria tend to inhabit the same individual mycorrhizal root-tip. Thus, rather than using statistical inference to estimate coefficients of association between taxa across samples, our study uses direct counts of root-tips on which a pair of fungi and bacteria/archaea are found, thereby relying on a fine-tuned experimental design to characterize the observed networks.

We constructed weighted bipartite networks of fungal and bacterial/archaeal communities of 62 mycorrhizal root-tips. The networks were calculated with relative abundance data. In the bipartite networks, each element of the adjacency matrix corresponds to the number of root-tips on which a given fungal and bacterial taxa co-exist. We then estimated the modularity of this bipartite network to determine the assemblies of bacterial taxa that tend to inhabit the same root-tip as a given set of fungi. In more detail, we constructed five bipartite networks, each considering 6632 bacterial OTUs and different relative abundance cut-offs for fungal OTUs per individual root-tip. As such we created five different networks that either considered only dominant fungi or also less abundant ones. Network "25%" included all fungal OTUs with a relative abundance higher than 25 percent (26 OTUs), network "10%" consisted of all fungal OTUs >10% relative abundance (40 OTUs), network "5%" included all fungal OTUs >5% rel. abundance (54 OTUs), network "1%" consisted of all fungal OTUs >1% rel. abundance (126 OTUs), network "0.1%" included all fungal OTUs >0.1% rel. abundance (416 OTUs). We then computed modularity in all the networks and chose three networks (network "10%", network "5%" and network "0.1%") for further analysis. We compared the resulting structure between the networks considering only the highly abundant fungi (networks with >10% or >5% relative abundance) with the most complete network including also other less abundant fungi (>0.1% relative abundance: network "0.1%"). No substantial changes among the three networks would suggest that mainly the highly abundant fungal OTUs shape the assembly of co-existing bacteria, while contrasting network structures will suggest that the consideration of rare fungi generate fungal-bacterial/archaeal community assemblies.

We propose an ecological framework to assess two non-alternative processes through which rare fungi may contribute to changes in the fungal/bacterial network structure. We assess the presence of each process based on whether consideration of rare fungi results in new modules in the more complete network, or in changes in taxa composition in pre-existing modules.

On the one hand, a clumped distribution of rare fungi in new modules (vs. an even distribution among pre-existing modules) will suggest the emergence (or not) of specific bacterial associations with rare fungi. On the other hand, the higher (or lower) the proportion of highly abundant fungi that remain in the same module in both networks, the lower (or higher) the effect of rare fungi-induced changes in the similarity patterns of bacterial associations between the highly abundant fungi. We used a Pearson's Chi-squared test for count data to assess whether rare fungi are evenly distributed across modules in the most complete network, and a t-test to assess whether the proportion of highly abundant fungi that keep sharing the same module in the most complete networks is significantly different from 1.

**Modularity**. Modularity identifies groups of nodes that tend to interact more with each other than with the network as a whole. Different algorithms can be used to evaluate multiple partitions of nodes and select the optimal one that maximizes modularity.

We determined modularity using the DIRTLPAwb algorithm[79] implemented in the function "computeModules" in the R package "bipartite"[80]. This algorithm maximizes weighted modularity in bipartite networks at a higher speed than other algorithms such as QuanBiMo, making it attractive for detecting the modularity of larger networks[79]. A maximum of 1000 MCMC steps with a tolerance level of $10^{-10}$ was used in 100 iterations, retaining the iterations with the highest likelihood value as the optimal modular configuration. We tested whether our networks were significantly more modular than random networks by running the same algorithm in 100 random networks, with the same linkage density as the empirical one, which served as null model[81]. Modularity significance was tested for each iteration by comparing the empirical versus the random modularity indices using a z-score test[82]. After testing the modularity of our network, we determined the number of modules[83] and identified the bacterial and fungal taxa ascribed to each module with the function "listModuleInformation" in the R package bipartite 2.0[80].

**Reporting summary**. Further information on research design is available in the Nature Portfolio Reporting Summary linked to this article.

## Data availability

The raw sequence data were deposited and released in the National Center for Biotechnology Information (NCBI) Short Read Archive under BioProject ID PRJNA778470. OTU tables, Metadata and Taxonomy files used in the analysis are deposited on figshare and available at https://doi.org/10.6084/m9.figshare.21276675.v1.

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

## Acknowledgements

This work was financially supported by the Austrian Science Fund (FWF) project DK plus [W1257-B20]. AMN was supported by Spanish Ministry of Science, Innovation and Universities (RTI2018-099672-J-I00). CK received funding from the European Research Council (ERC) under the Europeam Union's Horizon 2020 research and innovation program (grant agreement No 819446). SEM work was carried out at Core Facility Cell Imaging and Ultrastructure Research at the university of Vienna. The authors would like to thank Daniela Gruber for help in SEM preparation.

## Author contributions

D.W. and C.K. designed the study. M.D., R.G., F.S., D.M., W.M., S.G., J.W., V.M., carried out the experimental work under the supervision of D.W., C.K. and A.R. R.G. and M.D. did preliminary tests. S.G., W.M., J.W. and V.M. helped during sampling. M.D. and R.G. did SEM analyses under supervision of M.W. M.D carried out the PCRs with support of F.S., D.M. helped in processing of the sequences. MD did the sequence analysis with support of F.S. A.M.N. created the scripts for network analysis and conceived the conceptual framework. Network analysis was carried out by M.D. under supervision of A.M.N., C.K. and D.W. M.D. wrote the manuscript, supported by D.W., A.M.N. and C.K. All co-authors contributed to paper revision.

## Competing interests

The authors declare no competing interests.
