## [Peer Review File · Communications Biology]

Reviewers' comments:

Reviewer #1 (Remarks to the Author):

The manuscript is interesting and well written although some points should be clarified. The planned work is original offering novel information on the relationship between mycorrhizal fungi and bacterial communities that is a topic well addressed in the past.

In the first parts of the manuscript it seems that there are plants with a diverse mycorrhizal rate while instead it has been reported that it is similar. When the authors claimed about dominant and rare fungi are they referring at the ECMs only or also at the soil?

The authors have grown the plants in controlled conditions, but in which substrate? The natural one?

Can the authors add something about the role of some interesting bacterial communities? For example, are there some species/strains involved in drought tolerance? My suggestion is to strengthen a bit the functional part related to the found communities.

Reviewer #2 (Remarks to the Author):

The manuscript offers a fine-scale depiction of the microbial communities that are associated with young, ectomycorrhizal-colonized beech tree roots transplanted into a greenhouse environment. Comparing community compositions across soil habitats (i.e., bulk soil, rhizosphere, and colonized ectomycorrhizal roots), the authors showed that microbial compositions (archaea, bacteria, fungi) varied across habitats. Their evidence suggested selective forces at play between particular bacterial groups and the roots of young, ectomycorrhizal-colonized beech trees. By implementing network-based methods, the authors also showed that the network modularity of microbial taxa changes as a function of microbial complexity. Specifically, the addition of abundant and rare fungal taxa reorganized microbial networks, which resultingly changed the strength of bipartite connections. The sum of the presented data highlights the importance of holistic approaches when investigating plant microbiome assembly.

While this is an interesting paper, the study does have some limitations. Two of the main findings (that single EM roots contain many fungal taxa, and that EM roots harbor distinct bacterial communities compared with bulk soil) are fairly solid and interesting, but similar results have been reported elsewhere (e.g. citations on line 96 & 98). My biggest concern though is that the headline finding that rare fungal taxa in EM roots may also contribute to structuring interactions in the mycorrhizosphere is novel but less substantiated. I say this because this result derives primarily from creating network analyses with varying numbers of fungal OTUs. To my mind the fact that analyses with different datasets have different outcomes is not surprising and some type of robust null analyses is needed to demonstrate that this is not just a statistical artefact. Beyond that, network analyses are correlational and so it is difficult to infer underlying biological processes. Thus, network analyses are not strong evidence that these rare OTUs are involved in meaningful ecological interactions.

Additional Comments

Line 133: I had a hard time interpreting Figure 1 EM micrographs and the CARD-FISH. It wasn't easy to tell in panel B if the bacteria were on EM hyphae or the surface of the plant rootlet. Similarly, it was totally unclear to me what interpretation to derive from the CARD-FISH.

Line 150: It's odd that these EM fungi are not observed in bulk soil as these fungi produce hyphae that scavenge in the soil and often make up a large fraction of soil DNA.

Line 164: It seems like a homogeneity of variance test is also necessary as PERMANOVA cannot distinguish between a difference in location vs. dispersion, and the NMDS presented seem to show large differences in dispersion. This seems like it may also be somewhat of an issue for the study as there should be only 1 bulk soil sample and multiple EM root samples. It's not surprising that a bulk soil sample aggregating across the entire microcosm should be different than a single rootlet making up a discrete fraction of the microcosm.

Line 167: Figure 2A and 2B are identical. However, according to the figure legend, Figure 2A should be depicting bacterial/archaeal OTUs, and Figure 2B should be depicting fungal OTUs. The associated text

in lines 140-144 also suggests that the content in Figure 2B may be incorrect.

Line 172: How is 'close proximity' being defined, and how does this operational definition inform the experimental schema? It is unclear how close the selected colonized root tips were to one another and whether the distance between tips provided any insight into phylogenetic or functional composition.

Line 174: Change 'was' to 'were' to avoid a disagreement between the subject and the linking verb.

Line 212: What are they, taxonomically? I can't tell from Figure S6.

Line 222: >10% RA across the entire dataset, or in individual samples?

Table 1: Odd that the total number of bacterial OTUs doesn't change. In some network analyses a significance cutoff is used to determine network membership. Was that not used here so all OTUs are assigned a module?

Line 244: 'Orders' should be changed to 'guilds' for clarity.

Line 250: The authors should consider condensing Figure 4. Panel (a) has more than 60 fungal orders, which causes many of the color codes overlap closely. Aesthetically, reducing the number of represented taxa to those that are both emphasized in the text and display disproportionate reconfigurations (or are dominant taxa) would clarify the intended message of this figure. The figure legend could likewise reflect these groupings (e.g., Orders with a relative abundance \leq 'X').

Alternatively, Figure 4 could be moved to the supplementary data section.

Line 271: So the message here is that incorporating rare fungi causes the network structure to change? How would you show that is not just a statistical artefact? What is the null model for adding more species to a network analysis?

Line 284: How is this biological and not just statistical? Is the assumption that the 10% networks are not accurate b/c they omit taxa that are there?

Line 291-292: If the results from the t-tests indicate significant differences between the compared networks, then the p values should be less than 0.001 (not greater than 0.001).

Line 321: This is not surprising at all since the unit is a single root tip, correct? Individual roots with different EM (which have been shown to co-occur at small scales in many studies) would of course have vastly different "communities".

Line 331: I found the discussion of EM community assembly on roots to be somewhat dissociated from the main thrust of the manuscript. The study doesn't really get at competition or disturbance. I would either cut this or try to better integrate it with the results.

Line 343: Based on what evidence?

Line 445: The presented data does not support the notion that rare fungi can create new habitats. It may support the notion that rare fungi alter community membership, composition, and microbe-microbe interaction potentials. These could of course lead to new habitats (depending on the adopted definition), but these data do not capture habitat synthesis.

Reviewer #3 (Remarks to the Author):

In this work the authors characterized the microbial communities of ectomycorrhizal root tips from *Fagus sylvatica* trees collected from the field but raised in a greenhouse. Interestingly, instead of focusing their analysis in the more common OTUs, the authors applied analysis to isolate and highlight the role those rare fungi have in the shaping of the characterized microbial communities. Overall, the work is well planned, and the statistical methods are robust and support their results. Their discussion is well bounded and they avoid the temptation of stating directionality in their co-relations.

I am particularly pleased with the triangle diagrams because they clearly communicate their findings

I have only minor comments for the authors:

1.- Figures 2a and 2b are identical, it seems you mistakenly used the diagram that represents bacterial OTUS in both figures

2.- For the section starting in line 172 It is not clear from the methods section how the proximity

between micorrhizal root tips was assessed. Are the authors just referring to samples from the same tree? or the position of each root tip sampled from the same tree was registered? if the answer is the second alternative maybe a better description would be: "Mycorrhizal root tips from the same tree

3.- Please include more information on the greenhouse setup used to raise the plants. I suspect that it allowed for the collection of root tips without plant uprooting and this is important for the answer of comment 2.

Reviewers' comments:

Reviewer #1 (Remarks to the Author):

The manuscript is interesting and well written although some points should be clarified. The planned work is original offering novel information on the relationship between mycorrhizal fungi and bacterial communities that is a topic well addressed in the past.

We would like to thank the reviewer for their positive assessment and constructive comments.

1.1 In the first parts of the manuscript it seems that there are plants with a diverse mycorrhizal rate while instead it has been reported that it is similar. When the authors claimed about dominant and rare fungi are they referring at the ECMs only or also at the soil?

Reply 1.1: We are not sure if we fully understand the first part of this comment. It seems the reviewer was confused by our description of plant mycorrhizal colonization. Mycorrhizal colonization, i.e. the percentage of plant root tips colonized by ECM, was similar across plants, while within each plant, root tips were colonized by diverse fungal communities. As to the second part of the comment: we were referring to dominant and rare members of the fungal communities on ectomycorrhizal root tips, as we used the root tip data as the basis for the network analysis. A large part of the fungal taxa found on root tips, however, were also found in the soil, albeit with different proportions (Fig. 2b). We have added some clarifying sentences at line 166/167 and line 502 to remove potential ambiguities and improve clarity for both points mentioned by the reviewer.

Lines 163-167: „As expected, ectomycorrhizal and other symbiotrophic fungi tended to be associated more with mycorrhizal root-tips than with the other habitats, whereas saprotrophic and pathotrophic fungi tended to occur more in bulk soil and rhizosphere habitats (Figure 2d). However, a large part of the fungal taxa found on root-tips was also found in the soil (Figure 2b,d).“

Line 502: „Mycorrhization (i. e. the percentage of plant root-tips colonized by ectomycorrhizal fungi) was similar across all investigated root systems.“

1.2 The authors have grown the plants in controlled conditions, but in which substrate? The natural one?

Reply 1.2: Yes, we used as a substrate a mixture of the soil collected from the site where the trees were sampled (A-horizon, 4 mm sieved) and perlite (soil:perlite (8:1, v/v). We have modified this description in the revised version of the manuscript for improved clarification and also added the information that plants have been grown in a greenhouse under ambient sunlight and temperature for a year (lines 487–491):

„The trees were planted in a mixture of soil (collected from the tree sampling site, A-horizon, 4 mm sieved) and perlite (soil:perlite (8:1, v/v)^{35,61} and grown in a greenhouse under ambient (outdoor) sunlight and temperature for a year.“

1.3. Can the authors add something about the role of some interesting bacterial communities? For example, are there some species/strains involved in drought tolerance? My suggestion is to strengthen a bit the functional part related to the found communities.

Reply 1.3: We thank the reviewer for this interesting suggestion. We have added a couple of sentences discussing the potential functional roles of the associated bacteria in the revised version of the manuscript (lines 378–386), which now reads:

“Many bacterial strains have been reported to enhance ectomycorrhizal formation (refs 45,52,53) and enhance plant nutrition and health, however there is still a lack of knowledge about the role of non-easily culturable bacteria living in the mycorrhizosphere. In crop plant systems, such as soybean roots, rice roots or sugarcane rhizosphere, it has been reported that Burkholderiales, Streptomycetales and Rhizobiales act as plant growth promoting bacteria and are the core-responsive bacteria under drought conditions, mediating drought tolerance (refs 54–56).”

Reviewer #2 (Remarks to the Author):

The manuscript offers a fine-scale depiction of the microbial communities that are associated with young, ectomycorrhizal-colonized beech tree roots transplanted into a greenhouse environment. Comparing community compositions across soil habitats (i.e., bulk soil, rhizosphere, and colonized ectomycorrhizal roots), the authors showed that microbial compositions (archaea, bacteria, fungi) varied across habitats. Their evidence suggested selective forces at play between particular bacterial groups and the roots of young, ectomycorrhizal-colonized beech trees. By implementing network-based methods, the authors also showed that the network modularity of microbial taxa changes as a function of microbial complexity. Specifically, the addition of abundant and rare fungal taxa reorganized microbial networks, which resultingly changed the strength of bipartite connections. The sum of the presented data highlights the importance of holistic approaches when investigating plant microbiome assembly.

We also would like thank reviewer #2 for their positive assessment of our study, and their constructive comments.

While this is an interesting paper, the study does have some limitations. Two of the main findings (that single EM roots contain many fungal taxa, and that EM roots harbor distinct bacterial communities compared with bulk soil) are fairly solid and interesting, but similar results have been reported elsewhere (e.g. citations on lines 96 & 98). My biggest concern though is that the headline finding that rare fungal taxa in EM roots may also contribute to structuring interactions in the mycorrhizosphere is novel but less substantiated. I say this because this result derives primarily from creating network analyses with varying numbers of fungal OTUs. To my mind the fact that analyses with different datasets have different outcomes is not surprising and some type of robust null analyses is needed to demonstrate that this is not just a statistical artefact.

We completely agree with the reviewer regarding the necessity to compare the results of network properties with null models that account for the differences in matrix dimensions and interactions distributions found in different data sets. In fact, we did compare the results of network properties with null models. The estimates of modularity reported in the manuscript are always referred to the modularity estimated using a null model (as described in the methods section). In the new version of the manuscript we have pointed this out more explicitly (lines 238–240, 253, line 670). We built this null model randomizing 100 databases with exactly the same dimensions and linkage density as the empirical one, and used the same algorithm to estimate modularity. Then, the observed modularity is standardized by the mean value and standard deviation of the modularity expected by this model using a z-score. We would like to thank the reviewer for bringing up this important point and we hope to have clarified it now in the new version of the manuscript.

Beyond that, network analyses are correlational and so it is difficult to infer underlying biological processes. Thus, network analyses are not strong evidence that these rare OTUs are involved in meaningful ecological interactions.

We agree with the reviewer that it is difficult to infer biological processes from co-occurrence networks. It is correct that microbial co-occurrence networks usually rely on a statistical inference of association metrics based on pairwise correlations of taxa. However, in our study we took advantage of the fact that we measured fungal and bacterial communities on small and distinct physical units (i.e. mycorrhizal root tips). This allowed us – instead of inferring the network links based on pairwise correlations – to directly observe co-occurrence of fungi and bacteria at an ecologically meaningful spatial scale. Specifically, we created weighted bipartite networks by directly counting the number of root tips on which a given fungal and bacterial taxa co-exist. This allows us to assess which fungi and bacteria tend to inhabit the same individual mycorrhizal root tip. We think that, by relying on this fine-tuned sampling design rather than on statistical inference, our approach enhances the reliability that observed co-occurrences represent ecologically meaningful patterns. The presence of a structuring ecological pattern is also indicated by the strong non-random structure of the resulting network. This same approach (i.e. the use of co-occurrence at a meaningful ecological scale; e.g. Interpreting a pollination interaction based on the observations of a given insect on flowers of a given plant species or dispersal based on the observation of a bird species removing fruits from a given plant species), combined with network analyses, has been widely used to understand the complexity of ecological interactions. This approach was firstly used to study antagonistic interactions back in 1973 (May 1993, Cohen 1978, Cohen et al. 1990, Pimm 1982), and more “recently”, since 1987, (although remarkably after 2000), it has also been expanded to understand the architecture for mutualisms and positive interactions (Jordano 1987, Bascompte & Jordano 2007, Bascompte et al 2003, Olesen et al. 2007, Verdú & Valeinte-Banuet 2008, Montesinos-Navarro 2012). Therefore, we consider there is enough evidence to support that despite not characterizing interactions in an experimental set up, network analyses have been previously shown to be a useful and ecologically meaningful tool to address the complexity of potential ecological interactions.

To provide a clearer and more detailed explanation of our network approach we have added more text in lines 117–125 and to lines 598–611)

Cohen, J. E. (1978). Food webs and niche space. *Monographs in population biology*, (11), 1–189.

- Cohen, J. E. (1990). A stochastic theory of community food webs. VI. Heterogeneous alternatives to the cascade model. *Theoretical population biology*, 37(1), 55–90.
- May, R. M. (1973). Qualitative stability in model ecosystems. *Ecology*, 54(3), 638–641.
- Pimm, S. L. (1982). *Food webs*. Springer, Dordrecht.
- Jordano, P. (1987). Patterns of mutualistic interactions in pollination and seed dispersal: connectance, dependence asymmetries, and coevolution. *The American Naturalist*, 129(5), 657–677.
- Bascompte, J., & Jordano, P. (2007). Plant–animal mutualistic networks: the architecture of biodiversity. *Annu. Rev. Ecol. Evol. Syst.*, 38, 567–593.
- Olesen, J. M., Bascompte, J., Dupont, Y. L., & Jordano, P. (2007). The modularity of pollination networks. *Proceedings of the National Academy of Sciences*, 104(50), 19891–19896.

- Verdú, M., & Valiente-Banuet, A. (2008). The nested assembly of plant facilitation networks prevents species extinctions. *The American Naturalist*, 172(6), 751–760.
- Bascompte, J., Jordano, P., Melián, C. J., & Olesen, J. M. (2003). The nested assembly of plant–animal mutualistic networks. *Proceedings of the National Academy of Sciences*, 100(16), 9383–9387.
- Montesinos-Navarro, A., Segarra-Moragues, J. G., Valiente-Banuet, A., & Verdú, M. (2012). The network structure of plant–arbuscular mycorrhizal fungi. *New Phytologist*, 194(2), 536–547.

Additional Comments

2.1 Line 133: I had a hard time interpreting Figure 1 EM micrographs and the CARD-FISH. It wasn't easy to tell in panel B if the bacteria were on EM hyphae or the surface of the plant rootlet. Similarly, it was totally unclear to me what interpretation to derive from the CARD-FISH.

Reply 2.1: We thank the reviewer for bringing this to our attention and have made efforts to clarify Figure 1 and its interpretation (see arrows and text in Figure 1 below). Figure 1, panel B shows a coherent series of SEM images. Starting from roots with root-tips, we zoomed in on image details (indicated by the white square on each picture) to show bacterial colonization on the fungal mantle of the selected mycorrhizal root-tip. The mantle consists of a fungal sheath completely surrounding the root tip. Cross-section images of root tips from this experiment can be found in Mayerhofer *et al.*, 2021, Figure 3.

As the CARD-FISH image is not important for the presented results, we removed it from the supplementary information.

Mayerhofer, W. *et al.* Recently photoassimilated carbon and fungus-delivered nitrogen are spatially correlated in the ectomycorrhizal tissue of *Fagus sylvatica*. *New Phytologist* (New Phytologist, 2021). doi:10.1111/nph.17591

2.2 Line 150: It's odd that these EM fungi are not observed in bulk soil as these fungi produce hyphae that scavenge in the soil and often make up a large fraction of soil DNA.

Reply 2.2: The respective EM fungal orders were indeed also observed in the bulk soil and rhizosphere habitats, however to a higher relative abundance associated to mycorrhizal root tips. We adapted the statement in lines 160–163, 166–167.

2.3 Line 164: It seems like a homogeneity of variance test is also necessary as

PERMANOVA cannot distinguish between a difference in location vs. dispersion, and the NMDS presented seem to show large differences in dispersion. This seems like it may also be somewhat of an issue for the study as there should be only 1 bulk soil sample and multiple EM root samples. It's not surprising that a bulk soil sample aggregating across the entire microcosm should be different than a single rootlet making up a discrete fraction of the microcosm.

Reply 2.3: We thank the reviewer for pointing this out. The analysis of multivariate homogeneity (PERMDIST) was performed using the function `betadisper()` from the package `vegan` in R with the `bias.adjust=T` argument, which considers sample size differences. The dispersion between groups was not homogenous (`betadisper`, ANOVA, $F=21.929$, $p<0.001$), mostly due to the high group variance dispersion in mycorrhizal root tips samples, meaning that the PERMANOVA assumptions were not met and thus the results have to be interpreted with care. In more detail, pairwise comparisons of the permutation test for homogeneity of multivariate dispersions (with 9999 permutation) revealed that the dispersion of fungal and bacterial/archaeal communities on mycorrhizal root tip samples differed significantly from the dispersion of bulk and rhizosphere samples (16S rRNA and fungal ITS: `betadisper`, Permutation test, $p < 0.001$). Boxplots showing the significant differences in dispersion (visualizing the main distances from the group centroids) for fungal ITS and 16S rRNA sequences have been included as Supplementary Figure 5 (see below). The methods on the statistics have been included in Supplementary Methods 3. The information was included in the main text in lines 178–181.

Supplementary Figure 5

Boxplots based on tests of homogeneity of dispersion analysis representing main distances from group centroids for (a) fungal ITS sequence data and (b) 16S rRNA gene sequence data for bulk soil, rhizosphere and mycorrhizal root tips (MRT). P-values of pairwise comparisons of permutation tests for homogeneity of multivariate dispersions (PERMDIST run with 9999 permutations) are indicated.

2.4 Line 167: Figure 2A and 2B are identical. However, according to the figure legend, Figure 2A should be depicting bacterial/archaeal OTUs, and Figure 2B should be

depicting fungal OTUs. The associated text in lines 140–144 also suggests that the content in Figure 2B may be incorrect.

Reply 2.4: We apologize for the mistake. In the revised version of the manuscript, we now replaced Figure panel 2B with the correct figure depicting the fungal OTUs (see below).

2.5 Line 172: How is 'close proximity' being defined, and how does this operational definition inform the experimental schema? It is unclear how close the selected colonized root tips were to one another and whether the distance between tips provided any insight into phylogenetic or functional composition.

Reply 2.5: We apologize for the ambiguity in the previous version of the manuscript. The root system of one tree was split into about 6 parts and mycorrhizal root tips were picked from different root parts. The method section was adapted to better explain the sampling strategy of mycorrhizal root tips (lines 498–506). The dendrogram in Figure

3 shows that individual root tips sampled from the same tree did not necessarily cluster together based on microbial community composition, thus not necessarily harboring similar microbial communities. We have rephrased the header of this section to better reflect the focus of this paragraph (lines 187–188).

2.6 Line 174: Change 'was' to 'were' to avoid a disagreement between the subject and the linking verb.

Reply 2.6: “Was” has been exchanged to “were” (now line 189).

2.7 Line 212: What are they, taxonomically? I can't tell from Figure S6.

Reply 2.7: The four fungal OTUs that co-occurred on all root tips belonged to the fungal order Thelephorales and Agaricales. The 15 bacterial OTUs that co-occurred on all root tips belong to the bacterial orders Bacillales, Burkholderiales, Chthoniobacterales, Corynebacterales, Frankiales and Rhizobiales. The information has been included (in lines 228–233) for clarity. We have also included Supplementary Table 3 in the supplementary information stating the exact taxonomic classification of the co-occurring fungal and bacterial OTUs to improve clarity:

2.8 Line 222: >10% RA across the entire dataset, or in individual samples?

Reply 2.8: More than 10% relative abundance in individual samples. The information was included in lines 243, 245 for clarification.

2.9 Table 1: Odd that the total number of bacterial OTUs doesn't change. In some network analyses a significance cutoff is used to determine network membership. Was that not used here so all OTUs are assigned a module?

Reply 2.9 The total number of bacterial OTUs does not change because our approach consisted only of adding an increasing number of less abundant fungal OTUs to explore its effects on changes on the fungi–bacterial assemblages. As such, all bacterial OTUs were used as input in all calculated networks and only the number of fungal OTUs changed (according to the chosen cut-offs). We have adapted Table 1 (see below) to improve clarity and added columns that depict the number of OTUs used as input and after calculating the networks. After analysis, only network “25%” shows a different number of bacterial OTUs, which stems from the fact that at such a high fungal cut-off, not all bacteria are interacting with the present fungal OTUs (some bacterial OTUs interact with the fungi that were removed by the cut-off and are thus removed from the analysis).

Simulated annealing was used for module identification following Guimera, R. & Amaral 2005. This is a stochastic optimization technique that enables finding the configuration that maximizes the modularity of the network. Therefore, all the OTUs are assigned to a module in each iteration and the configuration that maximizes the modularity of the network is finally selected.

Guimera, R. & Amaral, L. A. N. Functional cartography of complex metabolic networks. *Nature* 433, 895–900 (2005).

a) Parameters for modularity of each network

Network	fungal OTU cut-off [% relative abundance]	Total OTUs after cut-off	z-Score	Modules	Total OTUs after analysis
"25%"	25	26 F, 6632 B	-3.55	5	26 F, 6622 B
"10%"	10	40 F, 6632 B	56.52	4	40 F, 6632 B
"5%"	5	54 F, 6632 B	95.47	6	54 F, 6632 B
"1%"	1	126 F, 6632 B	258.72	5	126 F, 6632 B
"0.1%"	0.1	416 F, 6632 B	851.24	4	416 F, 6632 B

b) Number of fungal and bacterial/archaeal OTUs in each module of the individual networks

Network	25%		10%		5%		1%		0.1%	
	F	B	F	B	F	B	F	B	F	B
	11	1835	10	2023	2	675	3	165	6	136
	4	1446	6	1091	13	1721	44	2287	268	2868
	2	469	4	1268	5	898	52	2389	126	3380
	5	1533	20	2250	17	1765	4	283	16	248
	4	1339			9	953	23	1508		
					8	620				
Sum of OTUs	26	6622*	40	6632	54	6632	126	6632	416	6632
Sum of modules	5		4		6		5		4	

* In the 25% network not all bacterial/archaeal OTUs are interacting with the selected fungal OTUs and were thus not used in modularity analysis.

2.10 Line 244: 'Orders' should be changed to 'guilds' for clarity

Reply 2.10: „Orders“ has been changed to „guilds“ (now line 264).

2.11 Line 250: The authors should consider condensing Figure 4. Panel (a) has more than 60 fungal orders, which causes many of the color codes overlap closely. Aesthetically, reducing the number of represented taxa to those that are both emphasized in the text and display disproportionate reconfigurations (or are dominant taxa) would clarify the intended message of this figure. The figure legend could likewise reflect these groupings (e.g., Orders with a relative abundance \leq 'X'). Alternatively, Figure 4 could be moved to the supplementary data section.

Reply 2.11: We thank the reviewer for the suggestion. In the revised version of the manuscript, Figure 4 has been adapted to improve clarity and to make the figure more aesthetically pleasant. The depicted taxa have been reduced to the dominant taxa, the colors reflect these grouping and have been reduced to avoid overlap. All taxa (both, fungal orders and bacterial phyla) with a relative abundance lower than 1% were condensed and are depicted in white. Please see new Figure 4 below.

Figure 4. Structure of the modules in the investigated networks.

Ring graphs visualizing (a) network “10%”, (b) network “5%” and (c) network “0.1%”. Each network shows the structure of its modules (modules are numbered) and the surrounding rings correspond to the taxonomic composition of the respective module. The inner ring represents the composition of fungal orders in each module, ectomycorrhizal lifestyles are depicted striped. The outer ring represents the distribution of bacterial phyla. All taxa with a relative abundance < 1% are condensed and depicted in white. The strength of the connection between the modules of each network corresponds to the thickness of the connecting black line.

2.12 Line 271: So the message here is that incorporating rare fungi causes the network structure to change? How would you show that is not just a statistical artefact? What is the null model for adding more species to a network analysis?

Reply 2.12: We hope we have clarified this point above in the extensive answer provided in response to the main comments of reviewer #2.

2.13 Line 284: How is this biological and not just statistical? Is the assumption that the 10% networks are not accurate b/c they omit taxa that are there?

Reply 2.13: We hope we have clarified this point above in the extensive answer provided in response to the main comments of reviewer #2.

2.14 Line 291–292: If the results from the t-tests indicate significant differences between the compared networks, then the p values should be less than 0.001 (not greater than 0.001).

Reply 2.14: We apologize for the mistake; the symbol has been corrected (lines 315 and 316).

2.15 Line 321: This is not surprising at all since the unit is a single root tip, correct? Individual roots with different EM (which have been shown to co-occur at small scales in many studies) would of course have vastly different "communities".

Reply 2.15: The text has been modified accordingly (line 346).

2.16 Line 331: I found the discussion of EM community assembly on roots to be somewhat dissociated from the main thrust of the manuscript. The study doesn't really get at competition or disturbance. I would either cut this or try to better integrate it with the results.

Reply 2.16: We agree with the reviewer. This section was removed to improve the flow of the discussion section in the manuscript

2.17 Line 343: Based on what evidence?

Reply 2.17: We detected dense bacterial colonization on the ectomycorrhizal mantle surface via scanning electron microscopy. The information was included in the text (now line 358).

2.18 Line 445: The presented data does not support the notion that rare fungi can create new habitats. It may support the notion that rare fungi alter community membership, composition, and microbe–microbe interaction potentials. These could of course lead to new habitats (depending on the adopted definition), but these data do not capture habitat synthesis.

We thank the reviewer for the comment and have rephrased this sentence accordingly (lines 468–472).

Reviewer #3 (Remarks to the Author):

In this work the authors characterized the microbial communities of ectomycorrhizal root tips from *Fagus sylvatica* trees collected from the field but raised in a greenhouse. Interestingly, instead of focusing their analysis in the more common OTUS, the authors applied analysis to isolate and highlight the role those rare fungi have in the shaping of the characterized microbial communities. Overall, the work is well planned, and the statistical methods are robust and support their results. Their discussion is well

bounded and they avoid the temptation of stating directionality in their co-relations.

I am particularly pleased with the triangle diagrams because they clearly communicate their findings

We are glad that the reviewer finds our study interesting and thanks for the positive and constructive comments.

I have only minor comments for the authors:

3.1.– Figures 2a and 2b are identical, it seems you mistakenly used the diagram that represents bacterial OTUS in both figures

Reply 3.1: We apologize for the mistake and have modified the figure in the revised version of the manuscript to contain the correct panel.

3.2.– For the section starting in line 172 It is not clear from the methods section how the proximity between mycorrhizal root tips was assessed. Are the authors just referring to samples from the same tree? or the position of each root tip sampled from the same tree was registered? if the answer is the second alternative maybe a better description would be: "Mycorrhizal root tips from the same tree

Reply 3.2: We apologize for the ambiguity in the previous version of the manuscript. The root system of one tree was split into about 6 parts and mycorrhizal root tips were picked from different root parts. The method section was adapted to better explain the sampling strategy of mycorrhizal root tips (lines 498–506). The dendrogram in Figure 3 shows that individual root tips sampled from the same tree did not necessarily cluster together based on microbial community composition, thus not necessarily harboring similar microbial communities. We have rephrased the header of this section to better reflect the focus of this paragraph (lines 187–188).

3.3.– Please include more information on the greenhouse setup used to raise the plants. I suspect that it allowed for the collection of root tips without plant uprooting and this is important for the answer of comment 2.

Reply 3.3. More information on the experimental set up has been included in the revised version of the manuscript (lines 484–506). We added the information that plants have been grown in a greenhouse under ambient sunlight and temperature for a year (lines 487–491). To collect mycorrhizal root tips, the plants had to be taken out of the soil and were carefully taken apart to collect root tips from the root system. The information has been adapted in lines 498–506.

Reviewers' comments:

Reviewer #1 (Remarks to the Author):

The manuscript has been improved following the reviewers' comments.

Reviewer #4 (Remarks to the Author):

The paper is a nice piece of information about the importance of the rare fungi for the bacterial-archaea composition and interactions.

I have several comments that will improve the manuscript:

Lines 119 to 125. please re-write the phrase "No substantial changes in modularity..." and please explain or be more precise of what you mean with "no substantial changes"

Figure 1. Provide the ID of the mycobionts of the most abundant morphotypes in the description of the figure.

Figure 2. Specify what is MRT. Use different colors for the figures 2c-e because is very difficult to observe the difference in all the tones of green.

Rewrite the sentence in red from the lines 159 to 163: "Fungal OTUs that occurred in..."

Line 167-168: "The patterns of OTU distribution was similar when considering all OTUs". This sentence is very confusing because it seems that you were the whole time talking about the general patterns. Please clear it.

Line 189-181: "This means the PERMANOVA assumptions were not met and the results have to be interpreted with care" that is part of the discussion, out of place in the results.

Description of Figure 3. It is not clear what you mean with "while the remaining rare TOUs are colored in grey". You should change the color of OTU_245 and all the colors near to gray. Also, it is very confusing "Mycorrhizal root-tips originating from the same tree are highlighted in the same color and the numbers refer to the box number of the plant". Please make a figure less complicated to understand.

Line 206: what do you mean that "contribute to the morphological structure of each individual root-tip"?

Supplementary Figure 5 did not show what is written in the Lines 210-214.

Table 1a: present in a better way the data of the columns "Total OTUs after cut-off" and "Total OTUs after analysis" F and B values. How is it possible that the bacteria in the "Total OTUs after cut-off" did not vary? It is also unclear why there is a difference of number of OTUs after the analysis

Table 1b: It is also very confusing why do you present a list of Fungi (11, 4, 2, 5, 4...) and Bacteria (1835,1446, 469, 1533, 1339...) of each module, explain

Figure 4. The graph is really good but the networks inside the rings are not informative, is very difficult to observe, I don't see the reason to include them in the same figure.

Figure 5 description, line 289: "... appear empty (i.e. M1 and M2 in a)" has to be correct to M1 and M4, take off "in a"

Line 293: rewrite the sentence "We show statistical evidence for two patterns..."

Line 309: what do you refer with "the most complete network"?

Line 331 "might be selected" I think that is too cautious for your strong results, I would say are selected.

Line 349: "Especially multi-host ectomycorrhizal fungi" it's referred about the fungi host bacteria or that these fungal species are generalist with trees?

Line 377-384: "Many bacterial strains have been reported to enhance..." the order of the ideas needs better redaction.

Line 422 to 426: It is unclear, maybe this will arrange it "if the shifts of bacteria/archaea composition are induced by changes in their associations with abundant fungi, we expect that abundant fungi from the same module in the more restricted network, would not grouped together in the most complete network" ??? (the same idea has to be improve in the description of Figure 6)

Lines 426-432: erase (figure 6 panel 2.2) in the line 4.28-4.29, and explain these two sentences in a better way.

Figure 6. It's very difficult to follow the graphic idea, maybe it would be easier to specify in a legend with colors, like Blue, red, green, circle with lines =comparison between networks...

Also, is It better to say it "clump" instead of "group"?

Lines 467-473: Very repetitive, be concrete

Line 501. Provide de % of mycorrhization

Reviewers' comments:

Reviewer #1 (Remarks to the Author):

The manuscript has been improved following the reviewers' comments.

Reviewer #4 (Remarks to the Author):

The paper is a nice piece of information about the importance of the rare fungi for the bacterial-archaea composition and interactions.

I have several comments that will improve the manuscript:

We would like to thank the reviewer for the positive assessment and constructive comments.

1. Lines 119 to 125. please re-write the phrase "No substantial changes in modularity..." and please explain or be more precise of what you mean with "no substantial changes"

We have re-phrased the sentence for clarification (line 120): "No changes in modular structure among networks that gradually consider less abundant fungal taxa..."

2. Figure 1. Provide the ID of the mycobionts of the most abundant morphotypes in the description of the figure.

We agree that it is interesting to know the ID of the morphotypes, and we grouped the root tips into morphotypes based on an initial characterization of color and shape of the hyphae. Our sequencing results then revealed A1 and A2 encompassed a diverse fungal community. The classification of the most abundant OTUs (>50% of the reads) revealed that A1 belonged to Thelephoraceae/Tomentella and A2 to Agaricales/Hebelomataceae. The root tip depicted in A3 was classified as Cenococcum geophilum based on its distinct morphology and sequence data.

We have added this information in legend 1 (line 153-156).

3. Figure 2. Specify what is MRT. Use different colors for the figures 2c-e because it is very difficult to observe the difference in all the tones of green.

The abbreviation MRT (mycorrhizal root tip) is now explained in the figure legend. The colors have been adapted for more clarity.

4. Rewrite the sentence in red from the lines 159 to 163: "Fungal OTUs that occurred in..."

The sentence has been adapted to (lines 160–163):

“Within the 100 most abundant fungal OTUs, OTUs with higher relative abundances on mycorrhizal root tips than in bulk soil and rhizosphere habitats belonged to fungal orders Thelephorales, Sebaciales, Pezizales, Agaricales.”

5. Line 167–168: "The patterns of OTU distribution was similar when considering all OTUs". This sentence is very confusing because it seems that you were the whole time talking about the general patterns. Please clear it.

We apologize for the lack of clarity. The beginning of the paragraph referred to the 100 most abundant fungal and bacterial/archaeal OTUs (lines 165 to 174). Later in the paragraph, we refer to results considering all OTUs (starting in line 170). We hope we have improved the clarity throughout the paragraph. The particular sentence (line 180–181) now reads:

“When considering all OTUs (Supplementary Figure 1), the patterns of OTU distribution were similar to the ones observed for the 100 most abundant OTUs.”

6. Line 189–181: "This means the PERMANOVA assumptions were not met and the results have to be interpreted with care" that is part of the discussion, out of place in the results.

The sentence has been deleted from the results and added in the discussion (line 381).

7. Description of Figure 3. It is not clear what you mean with "while the remaining rare TOUs are colored in grey". You should change the color of OTU_245 and all the colors near to gray. Also, it is very confusing "Mycorrhizal root-tips originating from the same tree are highlighted in the same color and the numbers refer to the box number of the plant". Please make a figure less complicated to understand.

We thank the reviewer for the suggestions. The barplot depicts the 60 most abundant OTUs in color, all other (less abundant) OTUs are now colored in white for improved clarity.

Each bar refers to one mycorrhizal root-tip and the numbers underneath the bars indicate from which tree the mycorrhizal root tip originated. If the mycorrhizal root-tip originated from the same trees, the numbers are highlighted in the same color. We have adapted the figure legend for clarification

Figure 3. Taxonomic assignment of fungal OTUs on individual mycorrhizal root-tips.

Relative read abundance (%) of fungal OTUs associated to the investigated 62 mycorrhizal root-tips samples. The 60 most abundant fungal OTUs are depicted in color; all other (less abundant) OTUs are depicted in white. The

dendrogram indicates clustering of root tips based on fungal community composition (based on Bray-Curtis dissimilarity). Each bar represents one mycorrhizal root-tip sample; the number underneath refers to the tree from which the root-tip originated. Same trees are highlighted in the same color.

8. Line 206: what do you mean that "contribute to the morphological structure of each individual root-tip"?

Each mycorrhizal root-tip was inhabited by one to three dominant ectomycorrhizal OTUs, which comprised more than 50% of the reads. It is likely that those highly abundant fungi mostly contribute to the formation of the fungal mantle tissue of root-tips, which subsequently shape the morphological structure of each individual root-tip.

Across all mycorrhizal root tip samples, we found 19 OTUs with a relative abundance >50% that likely form the fungal mantle tissue.

We clarified the sentence, which now reads (lines 217–220):

"We identified 19 highly abundant fungal OTUs (relative abundance > 50% on individual root-tips), which likely contribute to the formation of the mantle tissue and the morphological structure of each individual root-tip (labeled OTUs in Supplementary Figure 5)."

9. Supplementary Figure 5 did not show what is written in the Lines 210–214.

The seven OTUs prevalent in more than 90% of all root-tips are indicated in the top right corner of Supplementary Figure 5. Those OTUs show the highest relative abundance and highest prevalence across all investigated mycorrhizal root-tip samples. The sentence now reads (line 226):

"...(see OTUs in top-right corner in Supplementary Figure 5)."

10. Table 1a: present in a better way the data of the columns "Total OTUs after cut-off" and "Total OTUs after analysis" F and B values. How is it possible that the bacteria in the "Total OTUs after cut-off" did not vary? It is also unclear why there is a difference of number of OTUs after the analysis

The bacteria do not vary in the "Total OTUs after cut-off" because we only used the depicted cut-offs (25, 10, 5, 1 and 0.1%) for fungal OTUs. We constructed the networks considering all 6632 bacterial OTUs, and the fungal OTUs after considering different relative abundance cut-offs per individual root tips.

The difference in bacterial OTUs after analysis in network 25% stems from the fact that some bacteria only interact with the fungi that were removed when applying this cut-off. For clarity, we have now removed the column "Total OTUs after analysis" in Table 1a and have indicated the exception of reduced number of bacterial OTUs considered in the 25% network with an asterisk.

The table has been further adapted for more clarity.

a) Parameters for modularity of each network

Network	fungal OTU cut-off [% relative abundance]	Total OTUs after fungal cut-off*	z-Score	Modules
"25%"	25	26 F, 6632 B	-3.55	5
"10%"	10	40 F, 6632 B	56.52	4
"5%"	5	54 F, 6632 B	95.47	6
"1%"	1	126 F, 6632 B	258.72	5
"0.1%"	0.1	416 F, 6632 B	851.24	4

*Please note, that the numbers of bacterial OTUs were the same across all networks because we only applied a cut-off to fungal OTUs.

b) Number of fungal and bacterial/archaeal OTUs in each module of the networks. Each network (in columns) has a different number of modules (depicted in rows) and each module consists of a certain number of fungal (F) and bacterial/archaeal (B) OTUs.

	Networks									
	25%		10%		5%		1%		0.1%	
	F	B	F	B	F	B	F	B	F	B
Modules	11	1835	10	2023	2	675	3	165	6	136
	4	1446	6	1091	13	1721	44	2287	268	2868
	2	469	4	1268	5	898	52	2389	126	3380
	5	1533	20	2250	17	1765	4	283	16	248
	4	1339			9	953	23	1508		
Sum of OTUs	26	6622**	40	6632	54	6632	126	6632	416	6632
Number of modules	5		4		6		5		4	

**Please note that when considering the relative abundance cut-off of 25%, only 6622 bacterial/archaeal OTUs were used in the modularity analysis. This is due to the fact that some bacterial OTUs were not interacting with the fungal OTUs remaining when considering the 25% cut-off.

11. Table 1b: It is also very confusing why do you present a list of Fungi (11, 4, 2, 5, 4...) and Bacteria (1835, 1446, 469, 1533, 1339...) of each module, explain

Each network (in columns) has a different number of modules (see number of rows), and each of those modules (per row) consists of a certain number of fungal and bacterial/archaeal OTUs (i.e. B and F under each network). In Table 1b we represent how many OTUs were included in each module for each of the networks. We added the explanation in the table legend for clarification.

12. Figure 4. The graph is really good but the networks inside the rings are not informative, is very difficult to observe, I don't see the reason to include them in the same figure.

The networks are included in the graph in order to see that each module has a different network structure. We included the networks and would like to keep them in the figure, so that the reader can observe the difference in module structure at one glance and to connect the information in the rings to the underlying network structure.

13. Figure 5 description, line 289: "... appear empty (i.e. M1 and M2 in a)" has to be correct to M1 and M4, take off "in a"

Thank you for pointing this out. M2 has been corrected to M4. Now line 302.

14.Line 293: rewrite the sentence "We show statistical evidence for two patterns..."

The sentence has been adapted for clarity. Now lines 305–309.

"We observed two patterns in the structure of the networks that explain the effect of rare fungi on the fungal–bacterial/archaeal assemblies. We show statistical evidence for those patterns that relate to two non–exclusive ecological processes."

15.Line 309: what do you refer with "the most complete network"?

The most complete network refers to network "0.1%" that includes most of the fungal OTUs. The information has been added in the sentence, now line 323.

"...taxa belonging to the same module in the restricted networks did not share the same module in the most complete network (network "0.1%")."

16.Line 331 "might be selected" I think that is too cautious for your strong results, I would say are selected.

Might has been removed from the sentence, now line 344

17.Line 349: "Especially multi–host ectomycorrhizal fungi" it's referred about the fungi host bacteria or that these fungal species are generalist with trees?

Here, multi–host ectomycorrhizal fungi refer to fungi that have a broad tree host range and were found to colonize multiple trees. The information has been included in the sentence, now reading (line 361–363)

"Especially multi–host ectomycorrhizal fungi colonizing different host trees, such as members of the genera *Inocybe*, *Cenococum*, *Laccaria* and *Russulales*, are known to be strong competitors".

18.Line 377–384: "Many bacterial strains have been reported to enhance..." the order of the ideas needs better redaction.

We thank the reviewer for pointing this out and have restructured the paragraph (lines 386 to 398)

19.Line 422 to 426: It is unclear, maybe this will arrange it "if the shifts of bacteria/archaea composition are induced by changes in their associations with abundant fungi, we expect that abundant fungi from the same module in the more restricted network, would not grouped together in the most complete network" ??? (the same idea has to be improve in the description of Figure 6)

Thank you for the suggestion, we have incorporated most of it, and tried to simplify it a bit more, both in the text (line 435)and in figure 6.:

"if changes are induced by a split of the group of bacteria/archaea that were previously associated with the same highly abundant fungi, we expect that abundant fungi from the same module in the more restricted network would not be grouped together in the most complete network"

20.Lines 426–432: erase (figure 6 panel 2.2) in the line 4.28–4.29, and explain these two sentences in a better way.

The paragraph has been adapted for more clarity and now reads (lines 441–444):

"If neither of those processes occur, shifts between the networks will be minimized. No substantial changes to the network structure can be considered when rare fungi are evenly distributed across modules in the most complete network and the proportion of highly abundant fungi that keep sharing the same module is not significantly different from 1 (Figure 6, panel 2.2). No changes between the networks structure will occur when rare fungi are evenly distributed across modules in the most complete network and highly abundant fungi keep sharing the same module (Figure 6, panel 2.2)."

We prefer to keep the reference to the figure and panel for consistency, as we refer to the different parts of the panels in that figure along the whole previous paragraph to support our statements.

21. Figure 6. It's very difficult to follow the graphic idea, maybe it would be easier to specify in a legend with colors, like Blue, red, green, circle with lines = comparison between networks... Also, is it better to say it "clump" instead of "group"?

We thank the reviewer for the suggestion. We have adapted Figure 6 and the legend for clarity and included a legend explaining the parameters used in the conceptual framework.

Figure 6. Conceptual framework on processes shaping the effect of fungi on the bacterial/archaeal communities associated with ectomycorrhizal root-tips.

Morphological features of ectomycorrhizal root-tips provide distinct microenvironments for smaller microorganisms such as bacteria and archaea. Such morphological features, which select for certain bacterial/archaeal communities, can be shaped by only the most abundant fungi, but may also be influenced by the entire fungal community, including the less abundant ones. Depending on the influence of rare fungal taxa, fungal-bacterial/archaeal bipartite networks may show different structures if only highly abundant fungi or the whole fungal community are considered. Here we present a conceptual framework for interpreting such emerging differences in network structure: We assume that a shift in network structure after considering rare fungi can result from two non-exclusive processes: a) the emergence of specific bacterial associations to the rare fungi, and b) changes in the similarity patterns of bacterial associations with the highly abundant fungi. If changes are induced by the emergence of specific bacterial associations to rare fungi, these fungi will tend to be **grouped** into certain (new) modules in the network that considers all fungi (1.1 and 1.2, notice the grey module only present in the most complete network). If changes are induced because rare fungi alter the relationship between bacteria/archaea and abundant fungi (f.e. by a split of the group of bacteria/archaea that were previously associated to the same highly abundant fungi) we expect to see that fungi which shared the same module in the 'abundant fungi only' network will be in separate modules in the 'all fungi' network (1.1 and 2.1, notice the green module consisting of a mix of colors (red and blue) in the lower part of the chord diagrams). These two processes can also simultaneously affect the shifts in the network (1.1), and if none of them take place the shifts will be minimized (2.2).

Networks for 'only abundant' and 'all fungi' are represented in the example chord diagrams in the middle of each panel (upper half shows only abundant fungi, lower half shows all fungi, colours depict different modules, as in Fig.5), which visualize a potential difference in module structure. The ellipses are colored according to the colors in the chord diagram: blue and red refers to non-changing modules in networks; grey refers to new modules that emerge due to the consideration of rare fungi and only contain rare fungi; green refers to modules that emerge due to changes in the highly-abundant fungi. Black triangles: abundant fungi, grey triangles: rare fungi, black circles: bacteria/archaea.

22. Lines 467–473: Very repetitive, be concrete.

The summarizing paragraph has been adapted (lines 496–498).

"In summary, our network analysis demonstrated that rare fungi can alter the potential for microbe–microbe interactions, leading to novel fungal–bacterial community assemblies."

23.Line 501. Provide de % of mycorrhization.

The percentage of mycorrhization (estimates on average 85%) has been included in line 527.